# A new metoposaurid (Temnospondyli) bonebed from the lower Popo Agie Formation (Carnian, Triassic) and an assessment of skeletal sorting

Aaron M. Kufner [1,2]*, Max E. Deckman[3], Hannah R. Miller[2,4], Calvin So[5,6], Brandon R. Price[2], David M. Lovelace [1,2]*

1 Department of Geoscience, University of Wisconsin-Madison, Madison, Wisconsin, United States of America, 2 University of Wisconsin Geology Museum, Madison, Wisconsin, United States of America, 3 Department of Geology, University of Georgia, Athens, Georgia, United States of America, 4 College of Design, University of Kentucky, Lexington, Kentucky, United States of America, 5 Department of Biological Sciences, George Washington University, Washington, District of Columbia, United States of America, 6 Negaunee Integrative Research Center, Field Museum of Natural History, S Lake Shore Dr, Chicago, Illinois, United States of America

* akufner@wisc.edu (AMK); dlovelace@wisc.edu (DML)

## Abstract

Metoposaurid-dominated bonebeds are relatively commonplace in Upper Triassic continental deposits with at least ten monodominant, densely-packed bonebeds globally. The biostratinomy of several classic localities in India, North America, and Poland have been explored in detail; however, variability in methods and resultant conclusions point to the need for a more rigorous approach to understanding both the taphonomic and the ecological origins of metoposaurid-dominated bonebeds. Here we present the first monodominant metoposaurid mass mortality assemblage from the Late Triassic Popo Agie Formation and the stratigraphically lowest known record of several fauna from the Popo Agie Fm including the first occurrence of *Buettnererpeton bakeri* in Wyoming. We employ previously tested binning methods based on perceived hydrodynamic equivalence ("Voorhies groups") to assess pre-burial skeletal sorting. We suggest a simple counting and normalization method that avoids the inherent bias introduced by the interpretation of hydrodynamic equivalence of skeletal elements in taxa that lack actualistic experimental data. In contrast to other North American metoposaurid bonebeds, the sedimentology and skeletal sorting analyses of the Nobby Knob quarry support an autochthonous origin of this assemblage in a fluvio-lacustrine system with relatively little pre-burial sorting. Despite differences in underlying assumptions regarding the dispersal potential of specific skeletal elements, binning methods tend to follow similar trends regardless of framework used to assess different assemblages.

**Data availability statement:** All relevant data are within the manuscript and its Supporting Information files.

**Funding:** A David B. Jones Foundation grant awarded to DML helped fund the excavation of this material. The funders had no role in study design, data collection and analysis, decision to publish, or preparation of the manuscript.

**Competing interests:** The authors have declared that no competing interests exist.

## Introduction

Metoposaurid stereospondyls are common in Upper Triassic non-marine successions in mid to low latitude basins of Laurasia and mid latitude basins of east Gondwana. They are known from innumerable single occurrences and no less than ten monodominant bonebeds (Fig 1), several of which have been explored in detail. Romer [1] provided the first interpretation of a monodominant metoposaurid bonebed–known as the Lamy amphibian quarry–as a "drying pond" scenario. Subsequent workers presented evidence of transport of the skeletons from the site of death and suggested that the Lamy amphibian quarry is an allochthonous assemblage that resulted from overbank flooding [2–4]. Metoposaurid bonebeds range from completely disarticulated and hydrodynamically sorted elements [2,3,5–7], to uncommon partial articulation but still dominated by disarticulated remains [8,9], to several fully articulated skeletons [10]. Differences between individual metoposaurid size within these bonebeds vary from narrow [5,7,10–12] to broad size ranges [10,13,14]. The

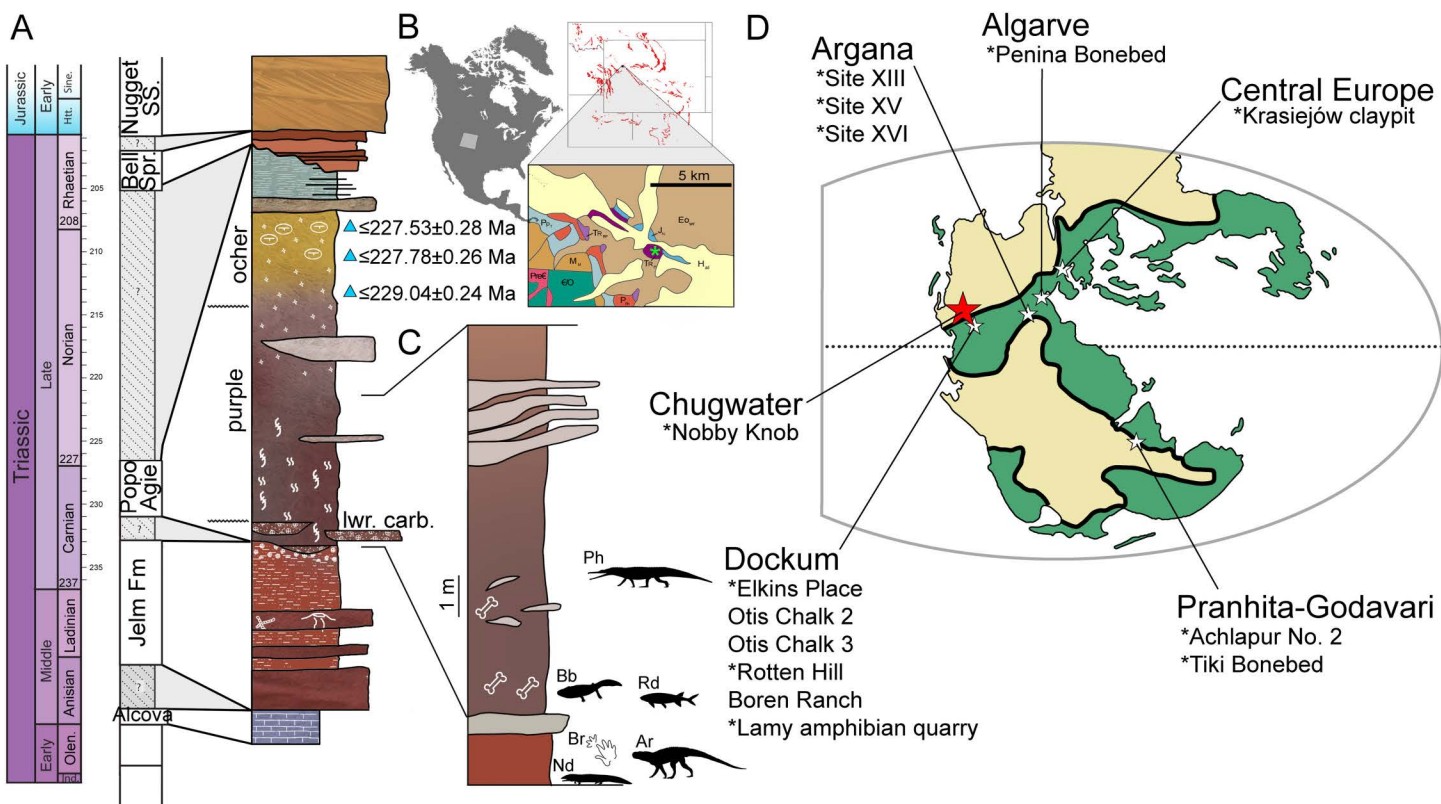

**Fig 1. Geological, temporal, and global context of the Nobby Knob bonebed.** (A) A generalized stratigraphic column of the upper Chugwater Group *sensu* [40] (triangles = CA-ID-TIMS detrital zircon ages [41]). (B) Geographic setting; first inset = areal extent of Chugwater Group, second inset = geological map (colors match the GTS) with the Nobby Knob locality denoted (red star). (C) A stratigraphic column of the local exposure of the Popo Agie Formation at the Nobby Knob locality. (D) Paleogeographic map with regional basins that contain Late Triassic metoposaurid-bearing bonebeds (* = monodominant metoposaurid bonebeds). Green portions of the map have more annual precipitation relative to tan (modified after [63]). Abbreviations: Ar, Archosauromorpha indet., Bb, *Buettnererpeton bakeri*, Br, *Brachychirotherium*, Nd, *Ninumbeehan dookoodukah*, Ph, Phytosauria indet., Rd, Redfieldiidae indet. Redfieldiid silhouette by AMK and archosauromorph silhouette by DML. *Chinlestegophis* (stand in for *Ninumbeehan*) silhouette in the public domain by T.M. Keesey. Other silhouettes used under Creative Commons Attribution 3.0 Unported https://creativecommons.org/licenses/by/3.0/ from phylopic.org. Metoposaurid silhouette after D. Bogdanov and phytosaur silhouette by S.A. Hartman. Geological map and outcrop extent of Chugwater Group data (B) reprinted from https://macrostrat.org API under a CC-BY-4.0 license. The North America silhouette and state outlines (B) based on 'USA_States_Generalized' and 'Territorial_Evolution_1867_-2003_TE_' shapefiles in the public domain (Sources: Esri; U.S. Department of Commerce, Census Bureau; U.S. Department of Commerce (DOC), National Oceanic and Atmospheric Administration (NOAA), National Ocean Service (NOS), National Geodetic Survey (NGS). The paleogeographic map outline of Pangea was modified after [64]. Paleoclimate map overlay modified after [65].

taxonomic composition of these bonebeds varies from monotaxic [5,7], to monodominant [2,3,10,11], to multitaxic [8,13,15]. Some metoposaurid-bearing bonebeds cannot be distinguished between a potentially time-averaged accumulation (e.g., Krasiejów clay pit, Rotten Hill bonebed, and Lamy amphibian quarry: [2,3,8,13]) and those considered to represent mass mortality events (e.g., Site XIII; [10]). Given the variability between modes of deposition and taxonomic composition, a more in depth understanding of the biostratinomic processes involved in the formation of metoposaurid-dominated bonebeds is required to interrogate the paleobiology of these long extinct fossil taxa.

Methods previously used to investigate the biostratinomy of metoposaurid-dominated bonebeds can be summarized into three categories: (1) paleocurrent indicators or proxies, (2) subaerial exposure indicators, and (3) skeletal sorting. Paleocurrent is typically measured through the alignment of primary sedimentary structures (e.g., ripple marks, imbrication, flute casts; see [16] for a summary). However, in the absence of primary sedimentary structures, some workers have used the hydrodynamic propensity of long bones to orient parallel to a unidirectional current to infer the presence and direction of flow ([17]; but see [18] for complications). Proxies for subaerial exposure can range from mudcracks in sediment to splintering or cracking of cortical bone induced by desiccation [5,19]. Subaerial or subaqueous exposure can also be indirectly inferred through the presence of bone modifications such as toothmarks of terrestrial or aquatic consumers [20]. Disarticulated vertebrate skeletons have been used to infer the degree of skeletal transport prior to final burial (e.g., [2,5,13,21,22]) similar to assessments of winnowing in sediments [23,24], microfossils [25], and shell-beds [26] inferred by distributions of grain size. Flume experiments with mammal skeletons allowed skeletal elements to be categorized into early, intermediate, and late dispersal groups [22]. Subsequently, workers have modified the dispersal group assignments of Voorhies [22] to assess the skeletal sorting of temnospondyl assemblages [2,13,27]. Flume experiments with non-mammal skeletons have demonstrated a continuum of dispersal potential that aligns with complexity of form and bone density and not strictly with homology as expected given the diversity and disparity of vertebrate skeletons [28]. In the absence of actualistic flume experiments for temnospondyls, the *a priori* assignment of temnospondyl skeletal elements to "Voorhies groups" is subjective and must be reassessed in an empirical framework of skeletal completeness based on known assemblages (e.g., [29]).

High density bonebeds are widespread in non-marine Upper Triassic localities of North America [1,6,13,30–35], India [5], Morocco [10], Europe [11,36], and South America [37,38]. All previously reported bonebeds from non-marine Upper Triassic strata of the western United States have been restricted to the Dockum Group and the Chinle Formation [2,13,15,30,33,34,39]. In contrast, vertebrate remains from the upper Chugwater Group (i.e., Crow Mountain, Jelm, and Popo Agie formations *sensu* [40]) are only known from isolated individuals, a few co-occurring conspecifics, or limited, disarticulated remains of a few taxa recovered from a single locality (see Table 1). Most published vertebrate fossils from the Popo Agie Formation are restricted to the ocher unit of the upper Popo Agie Fm and lower carbonate unit of the lower Popo Agie Fm [40] with only fragmentary remains published from the intervening purple unit (Table 1). Recent radioisotopic detrital zircon ages (Fig 1A; [41]) from the upper Popo Agie Fm place the purple unit within the Carnian Age (>229 Ma) of the Late Triassic which, along with the vertebrate fossil occurrences therein, provides a window into non-marine Carnian vertebrate communities of the western United States that are otherwise unknown or ambiguous in the south and southwest USA [40].

Here we report on the highly fossiliferous site Nobby Knob from the lowermost purple unit of the Popo Agie Formation that contains a bonebed we infer to be a monodominant

**Table 1. Fauna of the upper Chugwater Group of Wyoming represented by body fossils.** \*Minimum number of individuals determined based on personal observation. \*\*See nomenclatural note for reasoning behind renaming this site. \*\*\**Dolichobrachium* has not been formally considered a nomen dubium. Instead it has not been considered in the evaluation of "rauisuchians" since at least 1985 due to either being missing or disposed of due to damage beyond repair.

| Original Taxon/Specimen Number | Current Taxonomy | Locality | Unit | Minimum Number of Individuals (MNI) | Reference |
|---|---|---|---|---|---|
| *Anaschisma browni* | *Anaschisma browni* | Willow Creek(A) | OU | 2 | [42,43] |
| *Anaschisma brachygnatha* | | | | | [42,43] |
| *Koskinonodon princeps* | | Bull Lake Creek | OU | 5* | [44,45] |
| *Borborophagus wyomingensis* | | Sage Creek | OU | 2 | [44,45] |
| cf. *Borborophagus wyomingensis* | Metoposauridae indet. | Red Creek | OU | 1 | [44,45] |
| *Angistorhinus aeolamnis* | *Angistorhinus aeolamnis* | Attributed to Sage Creek | OU | 1 | [46,47] |
| *Paleorhinus parvus* | "*Paleorhinus*" (*Angistorhinus?*) *parvus* | Sage Creek | OU | v | [47,48] |
| *Angistorhinus gracilis* | Holotype lost | Baldwin Creek | OU | ≥1 | [47,49] |
| *Angistorhinus grandis* | *Angistorhinus grandis* | Between Baldwin Creek and \*\*Popo Agie Creek | OU | 1 | [47,49] |
| *Paleorhinus bransoni* | *Parasuchus bransoni* | \*\*Popo Agie Creek | OU | 1 | [50,51] |
| *Eubrachiosaurus browni* | *Eubrachiosaurus browni* | Little Popo Agie River | OU | 1 | [51,52] |
| *Brachybrachium brevipes* | Nomen dubium (? = *Eubrachiosaurus browni*) | Little Popo Agie River | OU | 1 | [51,52] |
| *Poposaurus gracilis* | *Poposaurus gracilis* | Little Popo Agie River | OU | 1 | [53,54] |
| *Angistorhinus maximus* | Holotype lost | Little Popo Agie River | OU | ≥1 | [47,48] |
| *Parasuchus* sp. | *Parasuchus* sp. | Ochre Hill | OU | 1 | [55] |
| *Dolichobrachium gracile* | \*\*\*Presumably lost or damaged beyond repair | Little Popo Agie River | OU | 1 | [48,51,56] |
| *Heptasuchus clarki* | *Heptasuchus clarki* | Clark/*Heptasuchus* | LC | 3 or 4 | [57,58] |
| "Single lungfish tooth" | Dipnoi | Clark/*Heptasuchus* | LC | 1 | [58] |
| "Smaller vertebrates" | Osteichthyes | Clark/*Heptasuchus* | LC | >1 | [58] |
| *Hyperodapedon* cf. *H. sanjuanensis* | *Beesiiwo cooowuse* | Willow Creek(B)/near Hole in the Wall | LC | 1 | [59,60] |
| UWGM 7027 and 7028 *Beesiiwo cooowuse* | *Beesiiwo cooowuse* | Cottonwood Creek | LC | 2 | [59] |
| UWGM 7029 | Hyperodapedontinae indet. | Cottonwood Creek | LC | 1 | [59] |
| TxVP 46035.1 | Hyperodapedontinae indet. | Cottonwood Creek | LC | 1 | [59] |
| *Unio* sp. | *Antediplodon* sp. and Unionoida indet. | "Popo Agie beds"/Nobby Knob | OU/PU | dozens | [61]; this study |
| UWGM 7264 | *Ninumbeehan dookoodukah* | Serendipity beds | Jelm Fm | dozens | [62] |
| UWGM 1995, 7549 | *Ahvaytum bahndooiveche* | Garrett's Surprise | PU | ≥1 | [41] |
| UWGM 7407, 7550 | Silesauridae indet. | Garrett's Surprise | PU | ≥1 | [41] |
| | **Previously Unknown Taxa** | | | | |
| UWGM 7578 | Parasuchidae indet. | "Above Nobby Knob" | PU | 1 | this study |
| See S1 Table | *Buettnererpeton bakeri* | Nobby Knob | PU | ≥19 | this study |
| UWGM 7575, 7579, 7586 | Redfieldiidae indet. | Nobby Knob | PU | ≥1 | this study |

metoposaurid mass mortality assemblage (Fig 1). We present a biostratinomic analysis of the assemblage and compare it with other metoposaurid-dominated bonebeds including two sites currently lacking detailed taphonomic analyses: the Elkins Place bonebed, a monotaxic metoposaurid bonebed with predominantly disarticulated remains from the lower Dockum Group of Texas, and Site XIII, a monodominant metoposaurid bonebed with articulated skeletons from the Timezgadiouine Formation of Morocco. In order to avoid binning biases based on perceived hydrodynamic equivalency or on homology

rather than actualistic experimental data, we use a simple counting and normalization method that can be used to assess skeletal sorting in historic collections lacking detailed site mapping data.

**Nomenclatural Note**: The historic sites "Sq*** Creek" and "Between Baldwin Creek and Sq*** Creek" [48] were named after a nearby creek that has since been renamed due to the derogatory nature of the original name. Places using the same pejorative on federal land in Wyoming have undergone name changes since 2021 [66]. The aforementioned creek was renamed to Popo Agie Creek, so we propose modifying the names of these sites in order to not maintain the use of this derogatory term. The two modified site names are: "Popo Agie Creek" to refer to the type locality of "*Paleorhinus*" *bransoni* and "Between Baldwin Creek and Popo Agie Creek" to refer to the type locality of *Angistorhinus grandis*.

**Institutional Abbreviations**: **UWGM**, University of Wisconsin Geology Museum, Madison, WI, USA.

## Materials and methods

### Stratigraphy

Three partial, but overlapping, trenched stratigraphic sections were measured in 2018 using a modified Jacob's staff, a compass, a laser pointer, and an early version of the iOS app "Jake for stratigraphy" [67]. Sites were chosen based on quality of exposure, distance from the Nobby Knob quarry, and accessibility (including a section measured directly through the NK quarry). Each section's composition was observed and recorded, including grain size, sedimentary structures, color [68], pedogenic features, and body/trace fossil occurrences. Based on characteristic compositions, stratigraphic bodies were grouped into several facies associations described in detail elsewhere [69].

### Specimen accessibility, preparation, and photography

During the summer of 2014, a field party from the University of Wisconsin Geology Museum (UWGM) collected the first *in situ* remains from the purple unit of the lower Popo Agie Formation (Chugwater Group) in Fremont Cty, WY (Fig 1) from a site named Nobby Knob (NK) referencing both the fictional *Discworld* character "Nobby" Nobbs as well as the knob-like topography of the outcrop. During subsequent field seasons, an increased effort to excavate the NK locality proceeded with excavations in 2016, 2018, and 2019. The NK fossils were prepared with contemporary methods of fossil preparation such as pneumatic oscillating air scribes and air abrasion with sodium bicarbonate. The fossils were consolidated and repaired when necessary primarily with paraloid B-72 and in some cases with Butvar B-98 if especially friable. Specimens were photographed with either a DSLR camera or a Dino-Lite Edge 3.0 USB Digital Microscope. Photographs were adjusted for color correction and focus stacking when necessary with Adobe Lightroom and Adobe Photoshop, respectively, and figures were compiled with Adobe Illustrator. A complete list of specimens prepared as of December 2024 is available in S1 Table. Field collections were conducted with permission of the Bureau of Land Management under permits PA16–WY–252 and PA16–WY–254.

Specimens from the Elkins Place (EP) bonebed were observed and counted firsthand for comparison due to clear skeletal sorting (see below) and sediment winnowing [6]. The EP locality is situated in the Camp Springs Conglomerate (also called the "Santa Rosa Sandstone") a high energy depositional system of the basal Dockum Group in west Texas [7,70]. In the literature, this unit has a regionally variable nomenclature; for internal consistency, we will refer to this unit as the Camp Springs Conglomerate *sensu* [70,71] throughout, though we will

note significant use of "Santa Rosa Sandstone" [72,73] or "Santa Rosa Formation" [71] where appropriate.

The NK specimens are reposited in the University of Wisconsin Geology Museum collection (S1 Table). The EP specimens are reposited in the University of Michigan Museum of Natural History Paleontology Collection (S2 Table). Specimens from Site XIII were counted via published photographic plates [10] and are reposited in the Muséum National de l'Histoire Naturelle collection (S3 Table).

## Taphonomy

A preliminary analysis of the taphonomy of the NK locality demonstrated little to no transport of skeletal elements in the NK locality [74], but here a more detailed analysis is undertaken. Three criteria are considered to evaluate hydrodynamic influence on the NK locality: (1) sediment grain size and distribution, (2) sedimentary structures or long bone orientation indicative of flow direction or regime, (3) hydrodynamic sorting of skeletal elements.

Metoposaurid remains from the NK, EP, and Site XIII localities were identified and categorized into Voorhies dispersal groups (VDG) following previous work (Table 2) based on the inferred surface area:volume ratio of temnospondyl remains contrasted with the mammalian remains originally tested [22]. To assess the validity of these VDG assignments, skeletal elements were counted, categorized into bins based on anatomical identification, and counts of elements were normalized to the expected number of skeletal elements based on the minimum number of individuals (MNI) for all localities. Due to a lack of complete articulated skeletons of North American metoposaurids, skeletal content was based on articulated specimens of *Dutuitosaurus* [10] and partially articulated specimens of *Metoposaurus krasiejowensis* [9,14]. The presented method assumes similar bone density across all skeletal elements which is rarely the case for air-breathing aquatic tetrapods [75], but in the absence of actualistic flume experiments for temnospondyls, we consider it sufficient for comparison with previous work.

Five portions of the NK locality in field jackets (NK16 J3 2-3, NK16 J8 1-2 [PL-011519], NK16 J10 3-5 [PL-015], NK18-D1-000 [NKAC], and NK19-C2-709.3 [NKAD]) were used to gather spatial data (S1–S6 Figs in S1 File). A quarry map was made for metoposaurid fossils collected in field jackets. Because elongate skeletal elements ($N = 207$, here defined as bones with length ≥ 4 times width following [2]) tend to orient with the long axis parallel to the direction of flow if a current is present during deposition [17], the azimuth of the long axis of each elongate, disarticulated element was measured. For all elements, the azimuth is bidirectional because it cannot be determined if either end of each bone would trend downstream or upstream. A Hermans-Rasson test and a Pycke test were performed to test for a multimodal distribution of the orientations of long bones using the functions "HR_test" and "pycke_test", respectively, in the R package CircMLE version 0.3.0 [76,77].

## Results

### Sedimentology and stratigraphy

Triassic stratigraphy of the upper Chugwater Group, Wyoming, includes the Jelm Formation which is unconformably(?) overlain by the mid-late Carnian-aged Popo Agie Formation. There is a roughly 25 Ma hiatus between the Popo Agie Formation and the overlying Bell Springs Formation (Upper Triassic), or more depending on regional stratigraphy (e.g., Jurassic-aged Gypsum Springs, Nugget Sandstone, or Sundance formations; [40,69]). The stratigraphy of the Popo Agie Formation is best known from the Lander area [69,78] where the lower carbonate, purple, ocher, and (locally) upper carbonate are well represented, but outcrops are visible along the entirety of the western slope of the Wind River Range.

**Table 2. Estimated skeletal content of *Buettnererpeton bakeri*, original Voorhies dispersal groups (VDG; dispersal potential groups: 1 = early; 2 = intermediate; 3 = late) of mammal skeletons [22], modifications for metoposaurid temnospondyls [2,3,13,21], and modifications for capitosaurid temnospondyls [27]. Numbers separated by a forward slash were considered intermediate between dispersal potential groups. \* Mandibles dispersed later if articulated in Voorhies' experiments [22]. \*\*Implicitly included as part of a larger element [22].**

| Element | Estimated Number per Skeleton (this study) | VDG (Voorhies, 1969) | VDG (Lucas et al., 2010, 2016; Rinehart et al., 2024) | VDG (Rinehart and Lucas, 2016) | VDG (Rakshit and Ray, 2020) |
|---|---|---|---|---|---|
| Skull | 1 | 3 | 3 | 3 | 2/3, 3 |
| Mandible | 2 | 2/3, 3* | 3 | 3 | 2/3, 3 |
| Intercentrum | ~40 | 1** | 1 | 2 | 1 |
| Neural Arch | ~94 | 1** | 1 | 2 | 1 |
| Hemal Arch | ~36 | NA | 1 | 2 | NA |
| Rib/Cleithrum | ~62 | 1 | 1 | 1 | 1 |
| Interclavicle | 1 | NA | 3 | 3 | 1/2 |
| Clavicle | 2 | 2 | 3 | 2 | 1/2 |
| Scapula (Scapulocoracoid) | 2 | 1/2 | NA | 2 | 1/2 |
| Humerus | 2 | 2 | 2 | 3 | 2 |
| Radius | 2 | 2 | 2 | 2 | 2 |
| Ulna | 2 | 1/2 | 2 | 2 | 2 |
| Metacarpals | 8–10 | 2 | 1 | 1 | 1/2 |
| Ilium | 2 | 2** | NA | 2 | 2 |
| Ischium | 2 | 2** | NA | 3 | 2 |
| Femur | 2 | 2 | 2 | 2 | 2 |
| Tibia | 2 | 2 | 2 | 2 | 2 |
| Fibula | 2 | NA | NA | 3 | 2 |
| Metatarsals | 10 | 2 | 2 | 1 | 1/2 |
| Phalanges | ~44–50 | 1/2 | 1 | 1 | 1/2 |

The larger scale stratigraphic architecture and geological context of the lower Popo Agie Formation between Lander and Dubois, WY, USA was described by Deckman and colleagues [69] where the authors identified 15 discrete lithofacies. Four of these facies are present in the lower Popo Agie strata associated with the Nobby Knob locality (Table 3) and are consistent with a distal splay facies association of a distributive fluvial depositional system [69].

In the Nobby Knob area there is a notable absence of conglomeratic facies typically associated with the lower carbonate unit [40,69]. As such, following the interpretation of Deckman and colleagues [69] we place the Jelm–Popo Agie contact at the base of the purple unit and the top of the underlying Jelm Formation (Fig 2) which regionally consists of paleosols with indicators of strong seasonality (e.g., redox mottling, prominent $B_k$ horizons; [69]).

The paleosol surface just below the purple unit at the NK locality has a prominent 30–50 cm thick carbonate nodule-rich horizon where nodules are partially coalesced forming a relatively resistant platform. The lower carbonate unit (considered absent at NK) is commonly expressed as a microconglomerate composed of reworked pedogenic carbonate clasts and minor amounts of vertebrate material [59]. We hypothesize that the paleosol surface at the top of the Jelm is the most likely source of the carbonate nodule clasts found in the conglomerates of the lowermost Popo Agie Formation. Although the lower carbonate is not present at NK, it has been observed at multiple locations within 1–5 km of the quarry. Although it is possible that differences in sedimentation distal to typical lower carbonate deposition in the Lander area are obscuring this otherwise prominent unit, we hypothesize that the carbonate nodule dominated horizon at the top of the Jelm represents a remnant

**Table 3. Lithofacies from the uppermost Jelm and lower Popo Agie formations.**

| Facies # | Name | Lithology, Bedding, and Color | Sedimentary Structures | Interpretation |
|---|---|---|---|---|
| 1 | Sandstone, horizontal lamination | Very fine-grained, well sorted, very thinly to thickly bedded sandstone. Red (2.5YR 5/6), yellowish red (5YR 5/6). | Planar lamination, parting lineation. | Upper flow regime planar beds deposited during unidirectional high energy flow. |
| 2 | Sandstone, current ripple cross stratification | Very fine-grained, well sorted, very thinly bedded sandstone. Red (2.5YR 5/6), yellowish red (5YR 5/6). | High angle, subcritical, small-scale, trough- and tabular- cross lamination. | Ripple formation under unidirectional flow. |
| 3 | Sandstone, massive | Massive, very fine-grained, well sorted, sandstone. Red (2.5YR 5/6), yellowish red (5YR 5/6), reddish brown (2.5YR 4/6), dark reddish brown (7.5R 3/3), dull yellowish brown (10YR 5/4). | Massive, structureless. | Very well sorted sands, inhibiting preservation of sedimentary structures, owing to similar grain size, poor preservation, or pedogenesis. |
| 4 | Mudstone, massive | Massive mudstone. Dark reddish brown (2.5YR 3/4), weak red (10R 4/3), reddish brown (2.5YR 4/6), dark reddish brown (7.5R 3/3), dull yellowish brown (10YR 5/4). | Massive, structureless. | Suspended sediment settling out in ponded areas or pedogenically modified mudstone. |

of the surface that is more commonly recycled into the higher-energy deposits of the lower carbonate unit of the Popo Agie Formation.

Locally, the majority of the lower purple unit is a pedogenically modified mudstone with disrupted primary sedimentary structures (e.g., facies 4, Table 3). At the NK locality the first 1–2 m of the purple unit consists of a silty mudstone that exhibits low angle crossbedding and > 1 cm fining upward layers of low-angle laminations (silty mudstone to mudstone). Between 2–4 m above the base, 0.25–1 m scale wedge-shaped peds with prominent slickensides, redox features, vertebrate remains, root traces, and minor amounts of plant material are common (Fig 2). From 5-7 m above the base of the purple unit, lateral accretion sets of very-fine to fine grained muddy-sandstone fill a channel form incised at least 2 m into mudstone deposits. This is consistent with other observations of intermixed fluvial sandstones and floodplain mudstone deposits regionally as well as the remainder of the NK section [69]. The upper Popo Agie Fm is present downdip of the NK locality (<100 m), but the top of the local "knob" ends at the purple-ocher transition.

## Taphonomy

**Spatial distribution of skeletal elements.** The majority of elongate disarticulated elements considered for the azimuth analyses were between 4–6 cm in length with the exception of mandibles disarticulated from skulls and a few isolated large ribs. These elongate bones show no preferred orientation (Fig 3) based on both the Hermans-Rasson test ($T = 1.477$, $P = 0.910$) and the Pycke test ($T = -1.772$, $P = 0.943$). There are several instances of articulation or close association, but the majority of the elements are disarticulated. A partial skeleton crosses the field jacket NK19-C2-709.3 with over one dozen nearly articulated intercentra, several articulated neural arches and ribs, and some girdle and limb elements with similar preservation and loose association with the axial skeleton (Fig 3; S6 Fig in S1 File). Additional elements found throughout the bonebed, including two partial, articulated hindlimbs (UWGM 7044) and multiple skulls, exhibit a steep and oblique orientation relative to the bedding plane. For instance, the hindlimbs are nearly vertically oriented, effectively "mired" in the sediment, and this orientation is more likely to occur in a low energy deposit [17]. There are also several ribs that plunge into the bonebed which has been previously interpreted as evidence for trampling in ceratopsian dinosaur bonebeds [79], although none of the bones are broken as is often the case in the dinosaur bonebeds. Three of the plate-like skulls are oriented obliquely to the horizontal plane of the bonebed, two of which are entirely perpendicular to the majority of the bones.

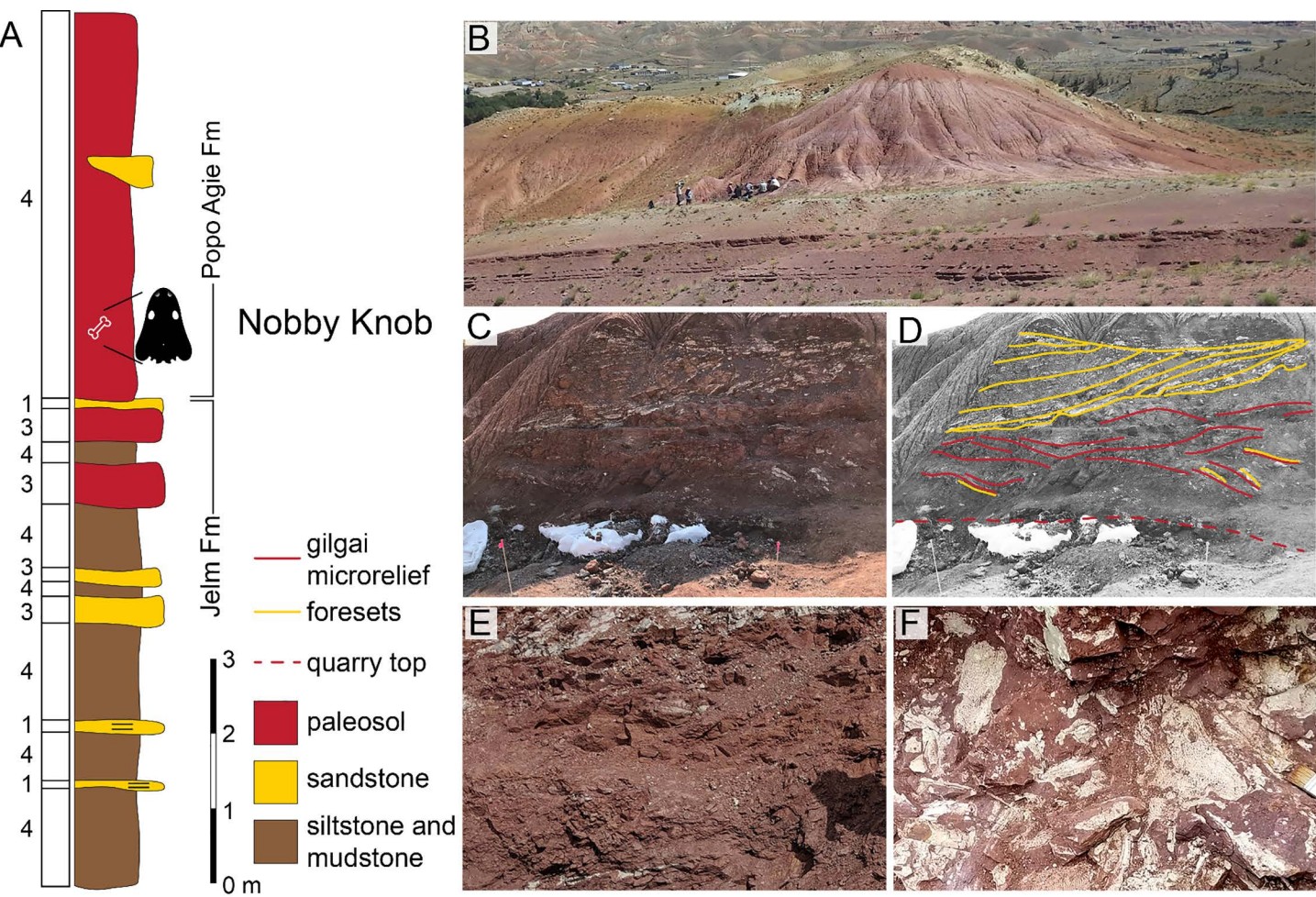

**Fig 2. Facies associations across the Jelm/Popo Agie Formation contact at the Nobby Knob locality.** (A) Facies associations 1–4 (see Table 3) of the uppermost Jelm and lowermost Popo Agie formations, Dubois, WY, USA (after [69]). (B) Photographs of the Nobby Knob outcrop (2014 field party for scale), (C) excavated cross section above the Nobby Knob quarry, and (D) interpretation of major bedding. Red dotted line marks the upper surface of the NK quarry, red solid lines = gilgai microrelief, yellow lines indicate fining-upward foresets of a fluvial channel. Note: channel cuts into underlying paleosol and the upper fluvial surface exhibits an erosional contact with overlying pedogenically modified mudstones. (E) Clay-rich vertic paleosols directly overlying (F) the bonebed layer which exhibits rhizoliths, redox mottling, and concentration of vertebrate remains.

At least two of these skulls have articulated mandibles, and at least one of those also has a partially articulated pectoral girdle.

**Sorting of skeletal elements.** The assignment of skeletal elements to VDG (Table 2) follows previous studies with some minor modifications to the counts to adjust for errors in limb element counts because the humerus and fibula were considered to sort differently from other limb elements by previous workers [2,13,21,22,27]. However, some flume experiments have demonstrated that the skeletal elements of freshwater turtles and alligators are not freed from the bed in discrete groups, but rather, the competent velocities (velocity of water at which the element is dislodged) of the elements lie along a continuum with "intermediate" dispersal elements overlapping in competent velocity with early and late dispersal elements [28,80]. This was similarly the case in Voorhies' original flume experiments on coyote and sheep skeletons where he assigned several elements to intermediate positions within his early, intermediate, and late dispersal group framework

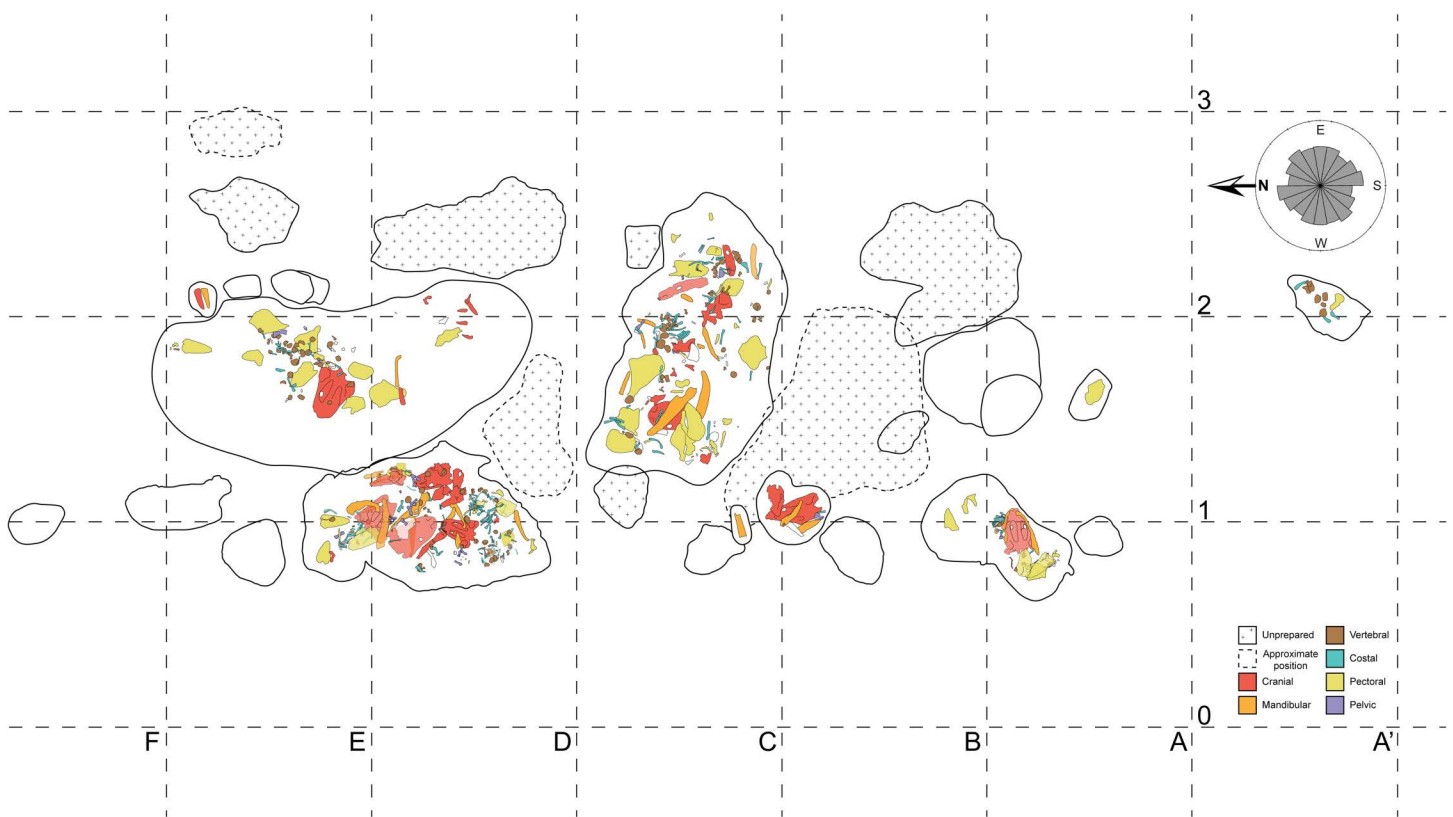

**Fig 3. Nobby Knob quarry map.** Individual metoposaurid elements were mapped from their respective field jackets. Note that all field jackets were prepared from the "field down" side, so the jacket maps are reflected to show their position from a "field up" view. Colors indicate anatomical identification as follows: red=skull roof, basicranium, or palate, orange=mandibular, brown=vertebral, blue=costal, yellow=pectoral girdle and forelimb, and purple=pelvic girdle and hindlimb. Field jackets with dashed outlines are in estimated positions, and those with crosses are unprepared. The rose diagram in the upper right indicates the orientation of disarticulated long bones (length ≥ 4 times width; *N* = 207). Detailed field jacket maps and the full list of metoposaurid elements from the NK locality as of December 2024 are available in S1 Table. Grid squares are 1 m².

[22]. Despite this, Blob [80] suggested that the use of discrete VDG was more practical than tabulating competent velocities for all skeletal elements. Some collection bias was introduced by a tendency of the field party to focus on jacketing the fragile, plate-like elements of the pectoral girdle and skull, however, all elements were considered both from field jackets and the surrounding trenches in the counts.

The MNI for EP is 13 based on 12 interclavicles of similar size and one large frontal among a few other exceptionally large non-duplicated elements. The MNI of the subset of Site XIII used in this study is 20 based on the maximum number of any individual skull roof element. The MNI of Nobby Knob is 19 based on 19 interclavicles and right clavicles. Skeletal elements normalized to the expected number of each element based on MNI revealed a pattern in each of the three bonebeds similar to what would be expected based on flume experiments of non-mammal skeletons [28,80]: a clear overrepresentation of low-profile, plate-like elements (e.g., skulls, interclavicles, and clavicles) and underrepresentation of diminutive skeletal elements (e.g., autopod bones, neural arches, and hemal arches). However, the skeletal elements of the NK locality do not exhibit evidence of significant hydrodynamic sorting (Figs 3 and 4) that would be expected if subjected to unidirectional flow in a fluvial system [17,22]. In contrast, the Elkins Place bonebed exhibits a strong overrepresentation of the flat plate-like skulls, clavicles, and interclavicles that

would be expected with the loss of earlier dispersing elements coincident with the absence of medium grained sandstone or finer sediment in the Camp Springs Conglomerate [6]. The figured blocks from Site XIII also exhibit overrepresentation of the skulls, clavicles, and interclavicles but with slightly better representation of the rest of the skeleton than the Elkins Place bonebed. However, this sample of Site XIII is not comprehensive and may not be representative of the entire locality.

Uniquely among metoposaurid bonebeds, the NK locality preserves articulated and disarticulated denticulate palatal plates (Figs 5E–G). Denticulate plates were embedded in the soft tissue of the palate in life [81] and have been reported from other metoposaurid assemblages [7,14] but have not been found in articulation within the palate. Disarticulated palatal plates have only been found underneath skulls, but articulated plates have been found in skulls with the palate facing up.

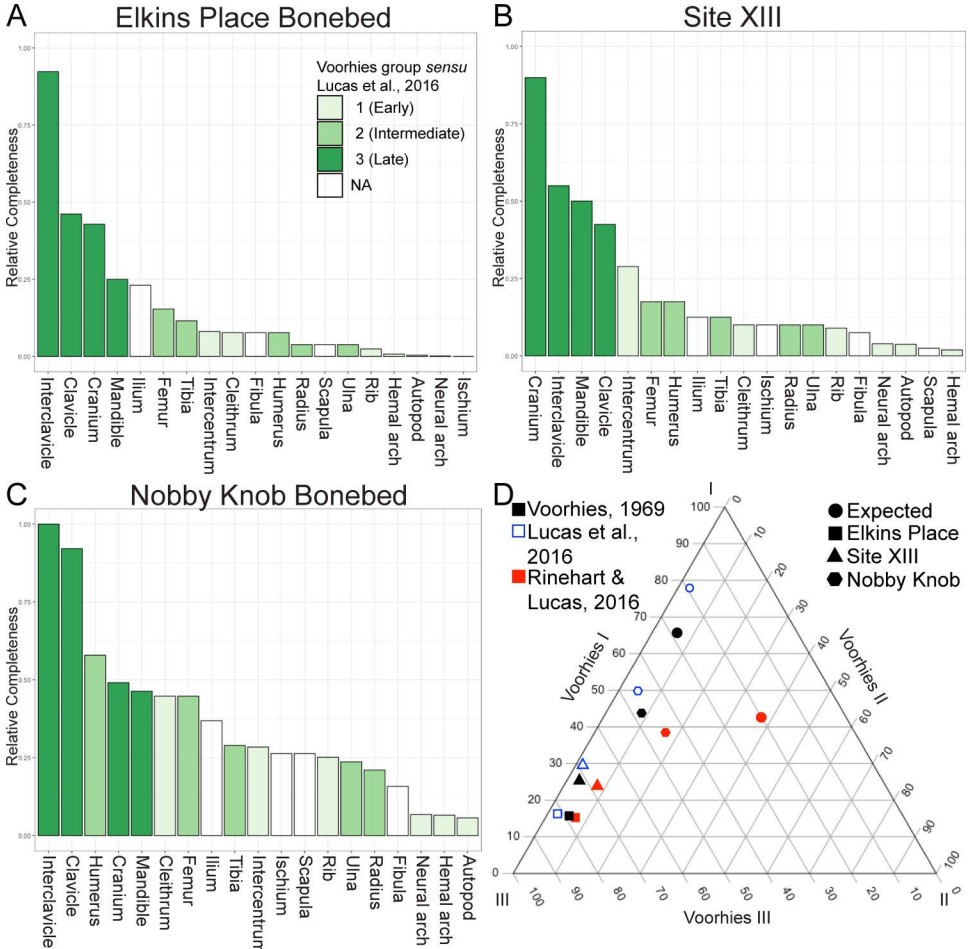

**Fig 4. Comparison of the relative completeness of metoposaurid skeletons between Elkins Place (TX, USA), Site XIII (Morocco), and Nobby Knob (WY, USA).** (A) Elkins Place elements normalized to expected value based on an MNI of 13. (B) Site XIII elements normalized to expected value based on an MNI of 20. (C) Nobby Knob elements normalized to expected value based on an MNI of 19. (D) Ternary diagram showing the expected proportion if all skeletal elements were present (circle) and the observed proportions for NK (hexagon), Site XIII (triangle), and EP (square) under different dispersal potential hypotheses (colors).

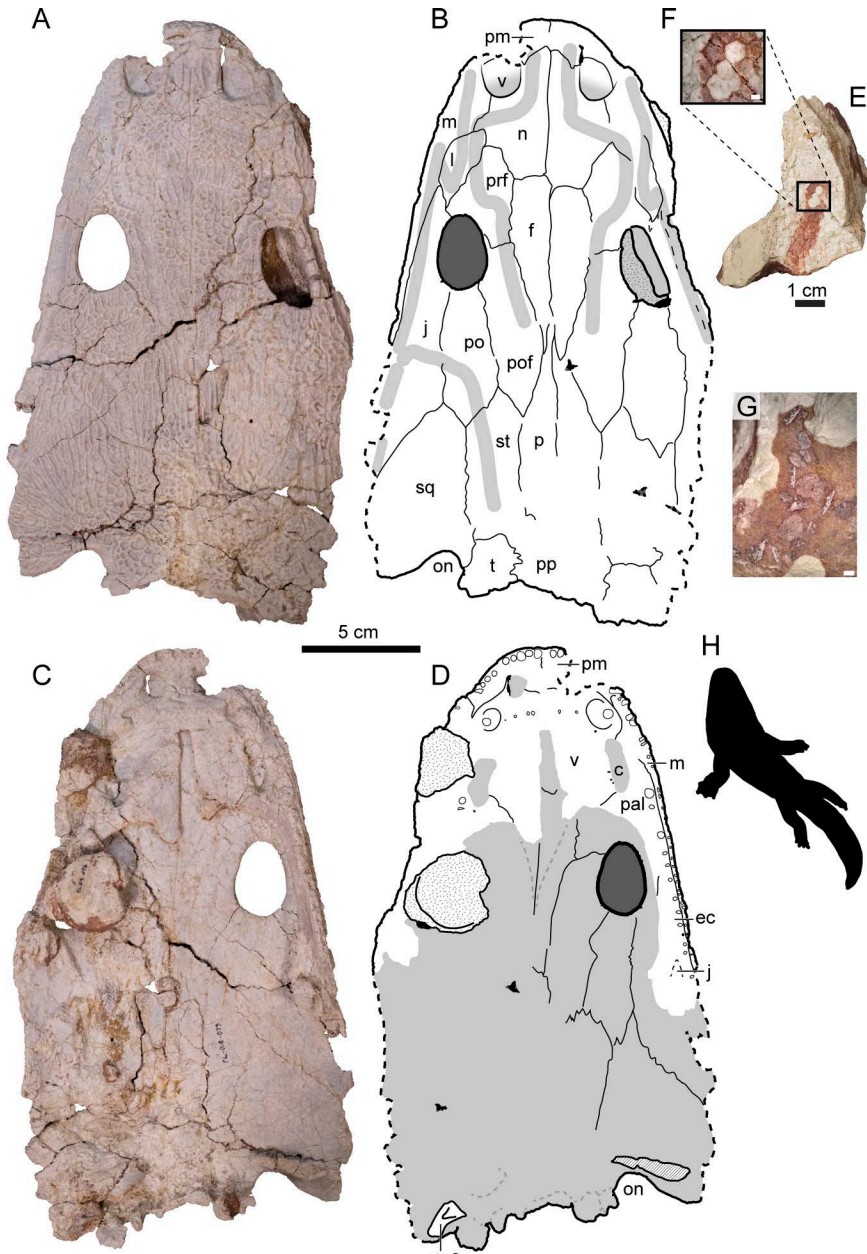

**Fig 5. Select specimens of *Buettnererpeton bakeri* from the Nobby Knob bonebed.** (A) Photograph of partial skull UWGM 7211 in dorsal view. (B) Interpretive drawing of the same. (C) Photograph of partial skull UWGM 7211 in ventral view. (D) Interpretive drawing of the same. (E) Photograph of articulated denticulate palatal plates in dorsal view UWGM 7574 and (F) close-up inset image of the same. (G) Photograph of disarticulated denticulate palatal plates associated with UWGM 7211. (H) Representative metoposaurid silhouette. Stippling represents non-cranial bones adhered to UWGM 7211; diagonal lines represent broken surfaces. Abbreviations: c, choana; ec, ectopterygoid; f, frontal; j, jugal; l, lacrimal; m, maxilla;, n, nasal; on, otic notch; p, parietal; pal, palatine; pm, premaxilla; po, postorbital; pof, postfrontal; pp, postparietal; prf, prefrontal; sq, squamosal; st, supratemporal; sta?, partial stapes?; t, tabular; v, vomer. Scale bar for A–D equals 5 cm, scale bars for E and G equal 1 mm, scale bar for F equals 1 cm. Metoposaurid silhouette by Adam J. Fitch used with permission.

**Bone modifications.** The majority of postmortem fracturing and deformation is unidirectional and consistent with sediment compaction and the shrink-swell cycle of surrounding clays. For example, the dorsal processes and laminae of several clavicles are broken and compressed against the ventral blade, and the posterior end of several skulls is splayed and flattened or sometimes mediolaterally compressed if lying on its side. Several bones are broken, or in some cases sheared along minor faults with slickensides (argillipedoturbation). One mandible has an unusual fracture pattern with a crushed glenoid region and the postsplenial split off from the overlying elements at about the mid length (UWGM 7568; S7 Fig in S1 File) differing from the typical unidirectional modification we attribute to sediment deformation. This could be due to trampling or other bioturbation, but it is an outlier within the context of the rest of the bonebed. No elements exhibit weathering or abrasion consistent with long term exposure at the surface or fluvial transport (e.g., [82]).

The Elkins Place (EP) bonebed from the Camp Springs Conglomerate (=Santa Rosa Sandstone or the Tecolotito Member of the Santa Rosa Formation) has commonly been interpreted as a fine to coarse sand-sized fluvial deposit interbedded with coarser grained conglomerates [6,70,72,73,83]. No site map of the EP bonebed exists so any comparison of spatial relationships between skeletal elements or orientation is impossible [7]. The representation of skeletal elements is biased toward plate-like bones of the skull and the pectoral girdle with very poor representation of autopodial elements and vertebral arches (Fig 4A). Many of the skull bones are disarticulated and only three skulls are essentially complete (with the exclusion of an essentially complete skull that was not observed firsthand housed at the Museum of Comparative Zoology: MCZ 1054), and there is no clear evidence of abrasion consistent with prolonged saltational transport on any bones [7,17,84].

**Bivalve preservation.** The mode of preservation of the bivalves (Fig 6) at Nobby Knob is unusual in that: (1) only the external morphology appears to be preserved (i.e., there are no internal molds/impressions, even in "butterflied" specimens), (2) there is no measurable thickness to specimens (e.g., shell material [nacre] is not present, nor a mold filled with secondary mineralization or preserved as a void), and (3) a dark "stain" is visible across the surface of some specimens which is mirrored on both halves when observed (usually after splitting sediment along a fracture plane during preparation or excavation of vertebrate material). Preliminary EDS analysis shows a miniscule calcium component (~0.6 normalized Wt%) along the preserved impression as well as the surrounding clay-rich matrix (S8 Fig in S1 File).

**Fossil occurrences.** *Invertebrate fossils*—The only invertebrates recovered from the NK locality are molds of unionoid bivalves (Fig 6). The molds exclusively preserve the external morphology on both the positive and negative relief (e.g., Fig 6A–B). Bivalve molds are found throughout the surrounding interval and were typically uncovered along fracture planes during excavation or preparation. The morphology of the bivalves is most consistent with species of the genus *Antediplodon* [85] *sensu* [86] (see S1 File).

*Vertebrate fossils*—The vast majority of vertebrates recovered from the NK locality are referable to the metoposaurid *Buettnererpeton bakeri* [87] *sensu* [7] (Fig 5). There are isolated fragments of an indeterminate redfieldiid actinopterygian typically recovered within the same layer as the metoposaurid material or just below this interval (Fig 7). A few shed archosauromorph teeth were found within the bonebed interval (Fig 8A–B), and a partial phytosaur mandible was recovered *ex situ* and can be stratigraphically constrained to within about 5 m above the bonebed (Fig 8D–J). See S1 File for extended descriptions and systematic paleontology of animal body fossils.

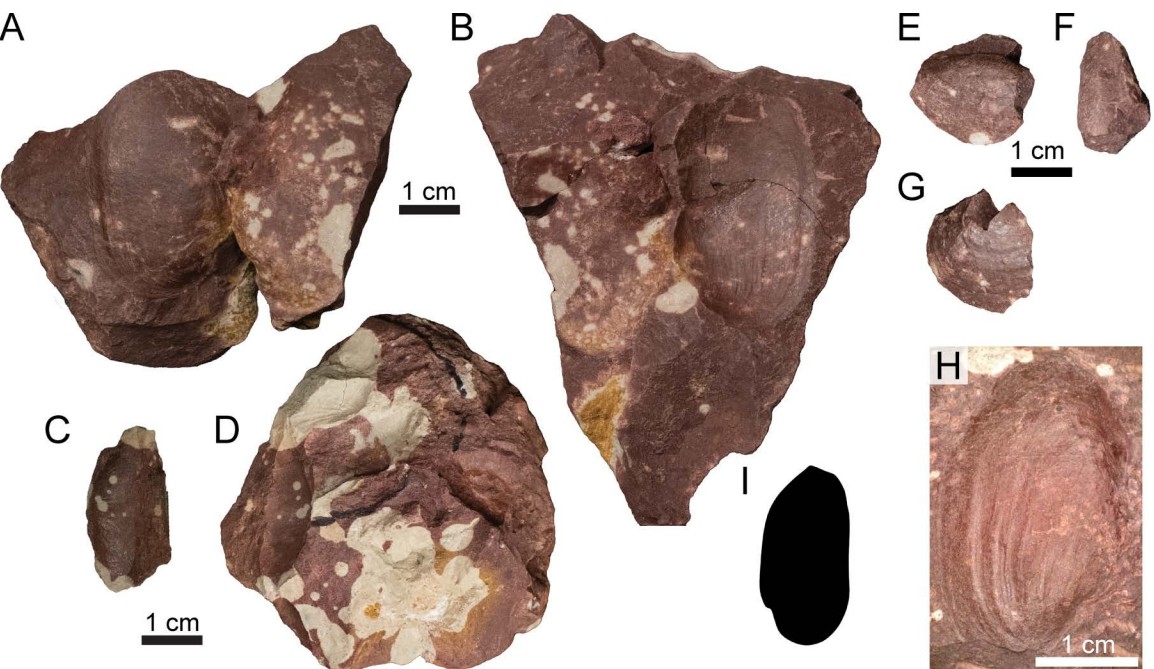

**Fig 6. Photographs of select unionoid bivalve molds (A–B, E–H cf. *Antediplodon* sp.; C–D Unoinoida indet.) from the Nobby Knob bonebed.** (A) UWGM 7571 part. (B) UWGM 7571 counterpart. (C) UWGM 7584 part. (D) UWGM 7584 counterpart. (E–G) partial external mold UWGM 7573 preserving the anterior left valve and the posterior right valve in (E) left lateral, (F) dorsal, and (G) right lateral views. (H) UWGM 7572. (I) Unionoid bivalve silhouette. Upper left scale bar for A–B, lower left for C–D, upper right for E–G, and lower right for H. Scale bars equal 1 cm. Silhouette by AMK based on UWGM 7572.

*Plant macrofossils*—Some plant remains have been recovered at the NK locality including abundant fragments of weathered petrified wood littering the surrounding area. Potential *in situ* plant remains are concentrated below the bonebed including possible leaves (Figs 9A, D–E), stems (Figs 9B–C), and roots (Fig 9F). Root traces are present throughout the interval as graygreen reduction zones, and some of these reduction zones contain carbonized root material (e.g., Fig 9F). Plant material is often not well preserved at the NK locality probably due to pedogenic overprinting of this interval. Previous reports of plant macrofossils from the Popo Agie Formation are restricted to the area around Lander, WY and the upper Popo Agie– the ocher unit or higher [61,90].

*Ichnofossils*—Despite the presence of isolated non-metoposaurid teeth among the metoposaurid remains, no modifications indicative of scavenging or predation (e.g., tooth or claw scores) have been identified. However, some isolated coprolites were recovered within the bonebed layer (Figs 9G–H). A lack of diagnostic features such as longitudinal striations or spirals precludes taxonomic identification. The coprolites are oblong and compressed in one direction resulting in an elongate oval cross section; this compression may be a taphonomic artifact based on similar compression of skeletal elements in this interval. Based solely on size and other taxa present, we propose that they were likely produced by tetrapods, but this is equivocal.

## Discussion

### Paleoenvironmental interpretation of the NK bonebed

Identification of the contact between the Popo Agie and Jelm formations is complicated by the absence of a carbonate- or quartz-grain conglomerate or a "chert pebble horizon" found

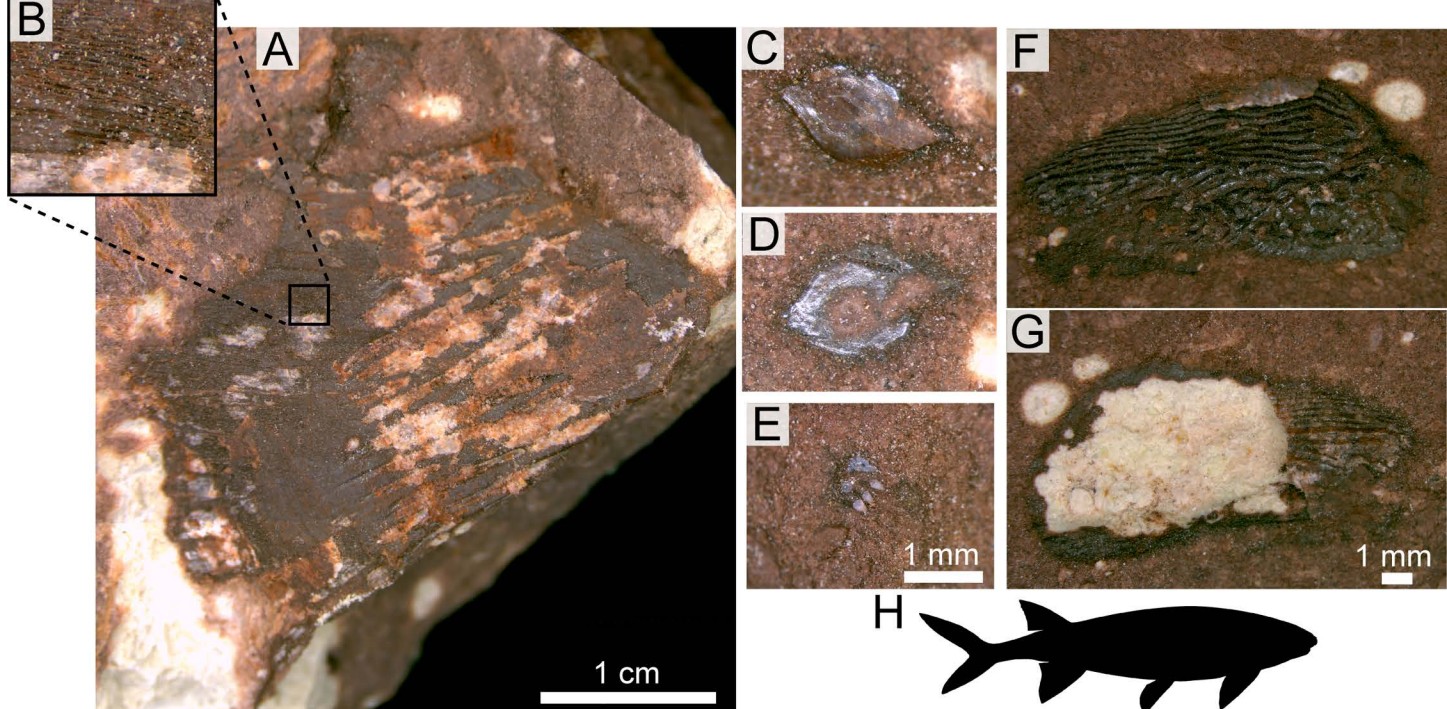

**Fig 7. Photographs of select indet. redfieldiid/actinopterygian body fossils from the Nobby Knob bonebed.** (A) UWGM 7576 possible actinopterygian fin with inset (B) showing elongate structures (=lepidotrichia?). (C) UWGM 7579 ganoid scale part. (D) UWGM 7579 scale counterpart. (E) UWGM 7586 tooth-bearing element (rostral?). (F) UWGM 7575 right supracleithrum counterpart. (G) UWGM 7575 partial right supracleithrum remaining part. (H) Representative redfieldiid silhouette. Scale bars equal values indicated in figure. Left scale bar for A, middle scale bar for C–E, and right scale bar for F–G. Redfieldiid silhouette by AMK based on *Lasalichthys stewarti* [88] *sensu* [89].

elsewhere in the Wind River Basin [91,92]. In the Dubois section we consider the contact to be above the Serendipity bed (Fig 2; [62]). Below this contact, pedogenesis is less common and fluvial-dominated deposits such as laterally accreting bars and splay deposits are abundant, although weak pedogenic overprinting is present throughout [69]. These crevasse splay deposits are similar in lithology to those found below the Channel Facies Association, indicating they may derive from a similar depositional system. Above the contact, paleosols are dominant, with few unaltered deposits. The thicker and more mature paleosols above the Serendipity bed indicate that there was much slower deposition in the region compared to downsection deposits.

The Nobby Knob bonebed is situated in pedogenically modified floodplain deposits interpreted as the distal portion of a larger distributive fluvial system [69]. Fluvial processes inferred from lateral accretion sets incised into floodplain deposits above and below the bonebed interval is the primary depositional agent. Fine laminations in the immediate interval surrounding the bonebed support a localized lower energy depositional system such as an oxbow lake or pond. Pedogenic overprinting partially obscures primary sedimentary structures, however remnant structures such as low angle fining upwards cross-bedding are consistent with an initial fluvial origin (e.g., point bar or splay deposits). Difficulties in determining lateral relationships along this interval of outcrop prevent a more detailed analysis. The Dubois area is considered to be dominated by Facies 4 (Table 1), with minor mud-rich splay deposits.

The overall interpretation of a fluvio-lacustrine depositional system for the lower Popo Agie Formation *sensu* [40] has not drastically changed since early work in the unit [93–95],

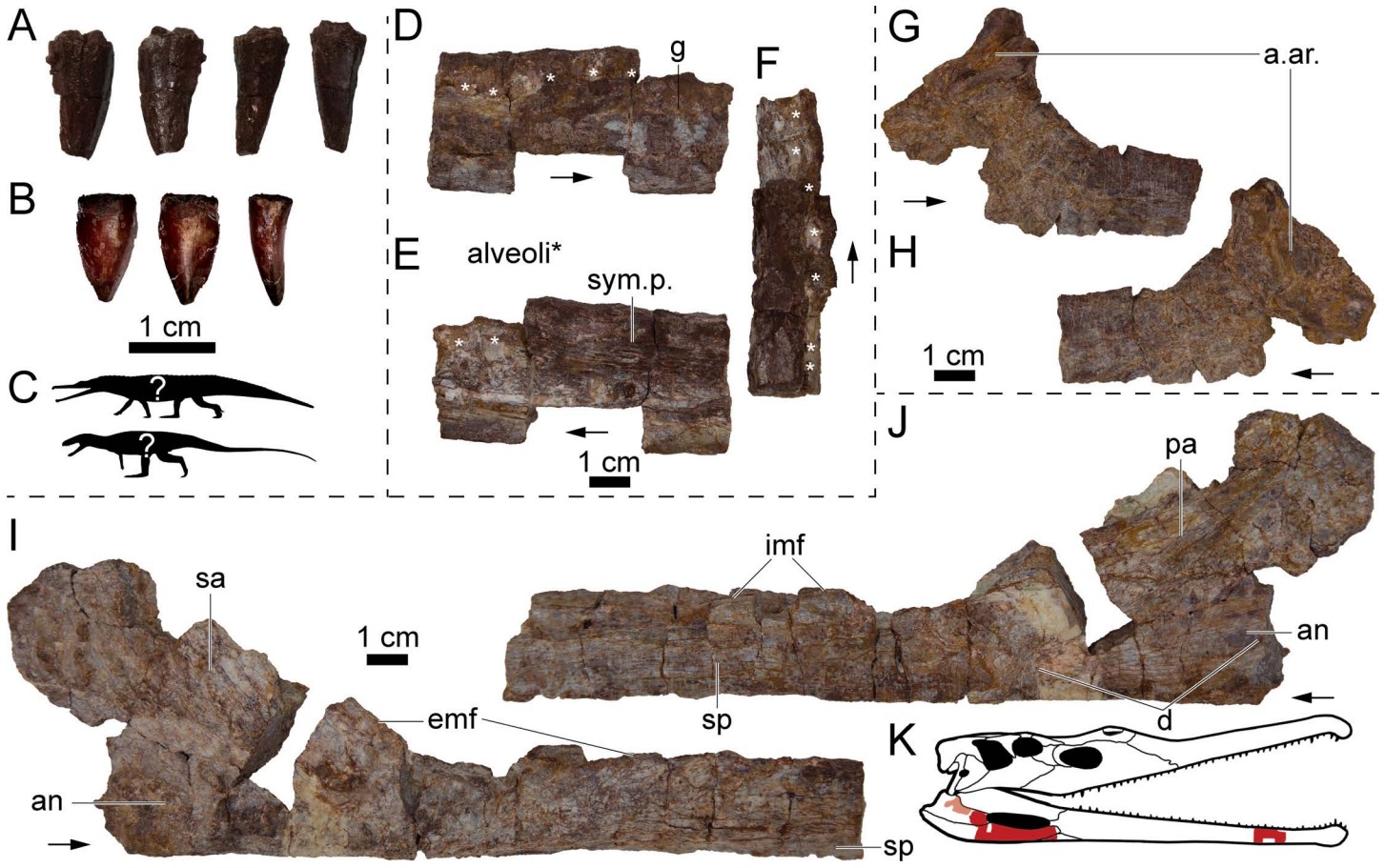

**Fig 8. Archosauromorph body fossils from the NK locality.** (A) Photographs of UWGM 7585 from left to right in lingual, labial, mesial, and distal views. (B) Photographs of UWGM 7569 from left to right in labial, lingual, and mesial or distal views. (C) Representative silhouettes of two archosauromorphs with serrated dentition known from the Popo Agie Formation, phytosaur (top) and poposaurid (bottom). (D–F) Photographs of partial right phytosaur dentary UWGM 1995 in (D) lateral, (E) medial, and (F) dorsal views. (G–H) Photographs of partial left(?) prearticular UWGM 7578 in (G) lingual(?) and (H) labial(?) views. (I–J) Partial right mandible UWGM 7578 in (I) labial and (J) lingual views. (K) Shaded parasuchid skull to show approximate positions of specimens. Alveoli in D–F marked with an asterisk (*). Abbreviations: a. ar., articulation with the articular; an, angular; d, depression; emf, margin of the external mandibular fenestra; imf, margin of the internal mandibular fenestra; pa, prearticular; sa, surangular; sym. p., symphyseal plate. Scale bars equal 1 cm. Phytosaur and poposaurid silhouettes by Scott Hartman used under Creative Commons Attribution 3.0 Unported https://creativecommons.org/licenses/by/3.0/ from phylopic.org. "*Paleorhinus*" skull illustration modified from original work by Scott Hartman used with permission.

although recent work rejects the hypothesis of a large persistent single lacustrine system as previously suggested for the upper Popo Agie Formation (e.g., [96]) in favor of smaller temporally variable lacustrine environments across the broader floodplain more common in medial and distal splay deposits of a distributive fluvial system [69]. In summary, our interpretation of the NK site is a low energy depositional system in a larger regional floodplain environment that incorporated a mass mortality event of unknown cause (Fig 10).

## The Elkins Place conundrum

Visual inspection of previously published images EP specimens containing residual matrix (figs 7, 10, & 18b: [7]) and outcrop images (see figs 2.10a, b, and d: [70]) of conglomeratic beds from the rock unit that has been referred to as the "Santa Rosa Sandstone" [70,72,73], clearly demonstrate very poor sorting, as noted by these and other authors [6,71,83]. The

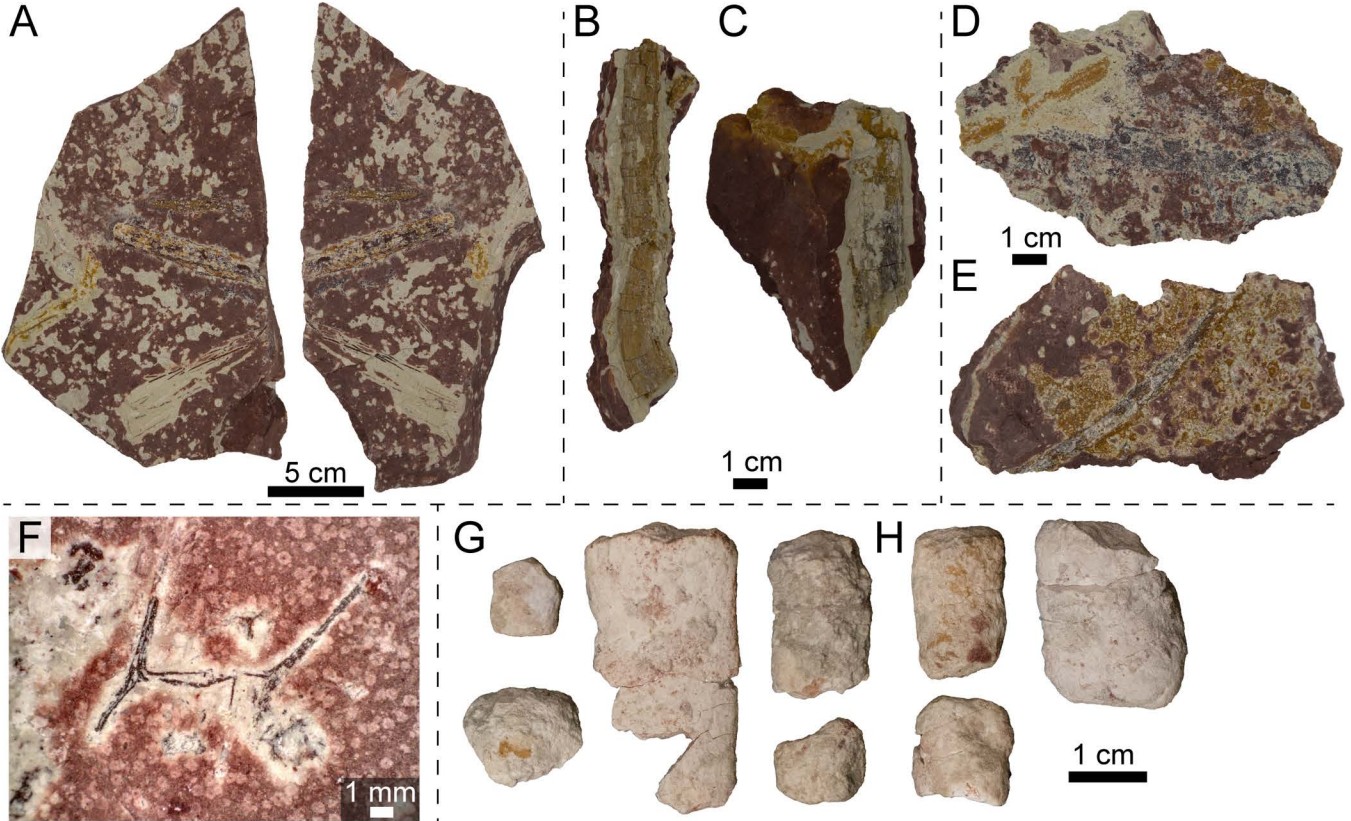

**Fig 9. Photographs of plant macrofossils and coprolites from the NK locality.** (A) Block with plant fragments, UWGM 7588. (B) Possible stem, UWGM 7580, in part. (C) Possible stem, UWGM 7581. (D) Plant material, UWGM 7587, in part. (E) Plant material, UWGM 7589, in part. (F) Root fossil, UWGM 7577, under cross-polarized light. (G) Coprolite fragments, UWGM 7582. (H) Coprolite fragments, UWGM 7583. Scale bar for A equals 5 cm, scale bars for B–E and G–H equal 1 cm, scale bar for F equals 1 mm.

larger quartz-rich (and bone) clasts are common but isolated and surrounded by a sand-sized matrix [6,7].

With the caveat that we lack detailed field observations of our own, we would like to point out that these matrix-supported clasts appear to be more consistent with debris (or possibly hyperconcentrated) flows than the more typical clast-supported conglomerates of braided streams they were inferred to represent. The matrix-supported "conglomeratic" deposits within the Camp Springs Conglomerate may be better explained by debris flow deposition which can follow local topography, occur both subaerially and subaqueously, and travel over short distances and low gradients [97–101]. Additionally, debris flows would explain the unusual deposits within the Camp Springs Conglomerate (=Santa Rosa Sandstone; or the Tecolotito Member of the Santa Rosa Formation) with large clasts supported in a sandy matrix (fig 2.10d: [70]). Entrainment in a debris flow during a flash-flooding event could explain the lack of abrasion and the presence of small (early dispersal; Voorhies Group I) and large (late dispersal; Voorhies Group III) elements in what has been previously described as high-energy conglomerates of a braided river system [73]. This would represent a depositional environment consistent with interpretations of a strongly seasonal climate [102]. The only other monodominant temnospondyl bonebed inferred to be hosted in a debris flow to our knowledge is an assemblage of *Eocyclotosaurus* from the Tecolotito bonebed within the Moenkopi Formation of New Mexico [27].

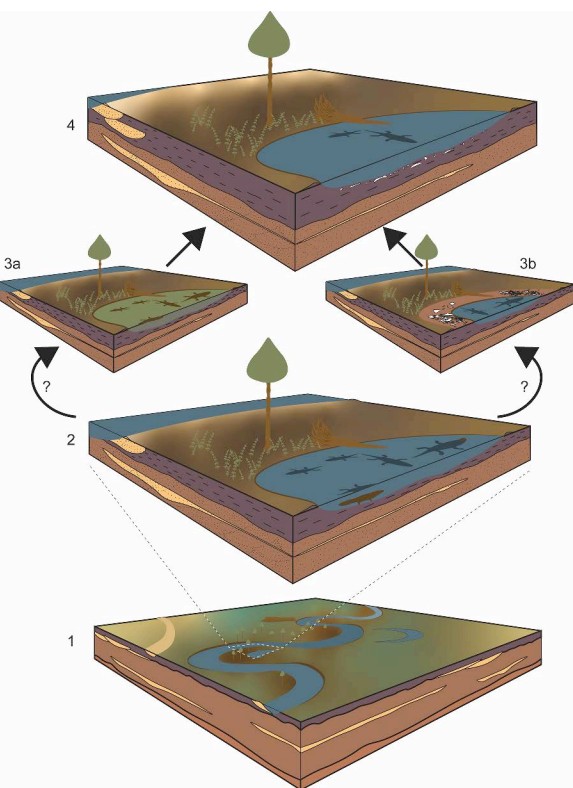

**Fig 10. Taphonomic scenario for the Nobby Knob site.** (1) Regional view of a distal floodplain river channel. (2) Local view of Nobby Knob site prior to mass mortality. (3a) Biotically induced mass mortality (e.g., eutrophication and subsequent oxygen depletion, disease, etc.). (3b) Abiotically induced mass mortality (e.g., drought). (4) Burial of metoposaurid remains as typical sedimentation continues.

## Bivalve preservation in a paleosol

The presence of unionoid bivalve molds and isolated redfieldiid bones (Figs 6 and 7) through the interval surrounding the bonebed provides additional support for a freshwater origin of the assemblage with low hydrodynamic influence, but the preservation of the bivalves is atypical. Although we do not have a clear explanation for the unusual mode of the observed bivalve preservation, we hypothesize that this phenomenon is a result of peri- or post-depositional dissolution of biominerals coupled with the higher initial preservation potential of the outermost organic layer. The very low concentration of measurable $Ca^+$ in the clay of both the mold and the surrounding matrix (S8 Fig in S1 File) is consistent with a lack of $B_k$ horizons in the Nobby Knob vertic paleosol and the absence of carbonate minerals. Although vertisols do not typically have a pH lower than 6.0 and more commonly between 7.0 and 8.5 [103,104], surface (or pore) water pH of 6–7.5 would be sufficient to dissolve the aragonitic biominerals within the outer prismatic layer and organic-mineral layered nacre [105] in pore waters undersaturated in calcium.

Dissolution of the outer prismatic aragonitic shell and inner nacre in a pH environment between 6–7.5 [105] would potentially leave behind organic structures and residues (i.e., periostracum and/or narcre-embedded organic layers [106]). We hypothesize the robust protein structure of the periostracum (composed of conchiolin) would effectively preserve the external morphology of the shell, once buried within the sediment, even after biomineral dissolution. Given the relatively uniform thickness of the periostracum, a mirror of that morphology

at the periostracum/prismatic layer boundary would be recorded as a negative. The loss of the biomineral portion of the shell would result in deflation of unlithified sediment through expansion and contraction (i.e., shrink-swell of clay component in vertic soils). The absence of measurable organic material (e.g., elemental carbon during energy dispersive spectroscopy analysis; see S8 Fig in S1 File) suggests subsequent diagenetic decomposition and removal of any organic residues [106]. Although, elemental carbon is at the limit of detectable spectra in the instrument used. It is unclear if the bivalves were transported or preserved as infauna at this time.

## Skeletal sorting and transport

The Nobby Knob bonebed exhibits a much lower degree of pre-burial sorting and, as a result, a higher representation of the skeleton than other monodominant North American metoposaurid bonebeds (Figs 4A, C; [2,3,13]). There is also no evidence for alignment of long bones with a current (Fig 5) unlike the Lamy amphibian quarry [2,3] and possibly the Rotten Hill bonebed [13]. Unfortunately, direct comparisons of skeletal sorting between the three presented sites and two previously studied North American metoposaurid bonebeds are not possible because either raw data on skeletal element counts are not provided or incomplete [2,3] or skeletal element counts were not done [13]. Outside of the late dispersal group, previously assigned Voorhies dispersal groups (VDG) for temnospondyls fit poorly with observed skeletal completeness in any of the bonebeds assessed here (Figs 4A–C).

Proposed VDG assignments of temnospondyls have relied on qualitative assessments of surface area:volume and complexity of morphology. One recent study used nearly the same dispersal potential groups proposed by Voorhies [22] despite the dissimilarity between many temnospondyl skeletal elements and their homologs in mammal skeletons [21]. Others have modified the VDG assignments of temnospondyl skeletal elements [2,3,13,27], however, some of the VDG assignments differ between these studies. For example, the metoposaurid clavicle has been treated as a late dispersal element (Voorhies group III) [2,3,13], but Rinehart and Lucas [27] considered the clavicle of the capitosaur *Eocyclotosaurus* to be an intermediate dispersal element (Voorhies group II) despite nearly identical morphology and thus surface area:volume. In addition, the relatively high density of metoposaurid dermal bones would be expected to increase competent velocity [107]. Regardless, there have been no actualistic experiments to determine the dispersal potential of temnospondyl skeletal elements.

Skeletal elements with similar shape such as skulls, interclavicles, and clavicles with large, flat regions tend to be represented in similar proportions (Fig 4). The humerus, fibula, and ischium were considered either intermediate or late dispersal elements in previous VDG applied to temnospondyls [2,3,13,21,27], but the representation of each of these elements in actual assemblages does not fit a late dispersal model but may fit an intermediate dispersal model (Fig 4). Diminutive skeletal elements such as vertebral arches, metapodials, and phalanges are poorly represented in all of the assessed bonebeds–in line with early dispersal and loss. The relative abundance of "intermediate" dispersal elements is highly variable between the three sites analyzed here, and it is not clear that this metric conveys any useful information to facilitate interpretation with respect to sorting or transport especially because "intermediate" dispersal elements overlap in competent velocity with early and late dispersal elements [22,28]. Interestingly, the estimated skeletal completeness from Site XIII with articulated skeletons appears more similar to the Elkins Place bonebed (Fig 4), however, this assessment is based solely on published photographic plates of blocks that may have been selectively favored for having the more diagnostic skulls and pectoral girdles [10]. The estimated relative skeletal completeness of the EP bonebed and Site XIII appears to follow a power curve, and

preliminary results from flume experiments show a similar trend in the competent velocities of skeletal elements in simulated dinosaur bones [108].

All parts of the skeleton of *B. bakeri* (except the epipterygoid) are represented in the NK bonebed including diminutive elements such as the distal phalanges and denticulate palatal plates (S1 Table). The underrepresentation of autopodial elements, neural arches, and hemal arches from the NK bonebed may be due to incomplete preparation of several of the quarry blocks since many of these elements are ~1 cm in maximum dimension. Preparation of NK material is ongoing and is expected to continue to yield more diminutive skeletal elements. The NK bonebed more than doubles the number of individuals of *B. bakeri* known to date and preserves a wide size range of individuals from a single site that can provide insight into the ontogeny of metoposaurids.

In addition to the presence/absence of skeletal elements, the alignment of elongate skeletal elements can be used to indicate the presence and direction of a current [2,17]. The NK bonebed exhibits no evidence of alignment of long bones (Fig 3). Articulation of metoposaurid elements is particularly rare in North America with only a few specimens previously reported [3,6,7,15]. Some skeletal elements from the NK bonebed are articulated including several skulls with mandibles, some of the postcranial axial skeleton, pectoral girdles and forelimbs, hindlimbs, and denticulate palatal plates. The presence of articulated denticulate palatal plates that would have been embedded in the soft tissue of the palate [81] implies the syndepositional presence of soft tissue in at least some individuals. The differences in palatal plate articulation between individuals could indicate differing degrees of decomposition prior to burial, or an effect of shielding by the plate-like skulls and pectoral girdles from post-depositional movement or shearing (i.e., shrink-swell, trampling or other bioturbation).

There are a number of large, plate-like elements that are preserved tangential to the bedding plane (oblique, and near-vertical alignment) positioned in such a manner that is rarely observed in the proposed low-energy depositional setting. We suggest these unique instances are most likely a result of localized or isolated specimens that were mobilized due to the shrink-swell actions of clay-rich soil prior to lithification. It is unlikely that these result from some form of hyperconcentrated flow considering the absence of any other matrix supported clasts (e.g., [109,110]) and the presence of fine laminations. It is also unlikely that their orientation is due to large tetrapod trampling given the lack of features typically associated with that scale of bioturbation (e.g., [79]).

The near monospecificity of the NK site together with sedimentological and taphonomic evidence of contemporaneous deposition suggest that the NK bonebed is the result of a biogenic aggregation of metoposaurids [84,111]. Several possible scenarios can result in biogenic aggregations of a single species such as breeding colonies [112], predator bone accumulation [113], or mass mortality [84,111,114]. No evidence of predation or scavenging is present on any of the metoposaurid remains and no osteological proxies for temnospondyl breeding are known. A monsoonal climate has been proposed for the underlying upper Jelm Formation resulting in seasonal aridity that would have shrunk and possibly dried-up isolated bodies of water [62]. The more developed paleosols of the lower purple unit indicate a more humid hydrologic cycle with higher water table relative to the underlying Jelm Fm, but the presence of vertic features such as gilgae topography and slickensides, as well as variable redoximorphic features (see Fig 2) indicate periodic (seasonal?) saturation and desaturation [115]. The remains of non-metoposaurid vertebrates within the bonebed interval appear to be incidental and possibly time-averaged or transported (e.g., disarticulated ray-finned fish remains and shed archosauromorph teeth). We conclude that the concentration of metoposaurids at this locality is likely the result of biogenic aggregation and plausible scenarios observed in modern amphibians include: seasonal breeding behavior and subsequent die-off [112], ontogenetic

niche partitioning [116,117], or anatomical and/or physiological limitations that prevent exodus from a drying body of water (e.g., [118]).

## Comparison with other metoposaurid sites

The Nobby Knob bonebed (NK) is the first unequivocal metoposaurid mass mortality assemblage from the Carnian-aged lower Popo Agie Formation. Two additional sites from the upper Popo Agie Fm are monodominant or monotaxic metoposaurid bonebeds that could be interpreted as mass mortality assemblages *sensu* Rogers et al. [84]: Bull Lake Creek and Willow Creek. The stratigraphic position of Bull Lake Creek is uncertain, but it is from the upper Popo Agie Formation and may be situated as low as the purple-ocher transitional zone [44,45]. The skeletal elements from Bull Lake Creek are from no less than five individuals and are almost entirely large, flat elements such as skulls, mandibles, interclavicles, and clavicles with the exception of one ilium and one intercentrum [45]. The Willow Creek locality is located within the ocher unit based on residual matrix on the holotype skull of *Anaschisma browni*, a referred skull (the holotype of "*Anaschisma brachygnatha*" [42]), and a referred mandible [43]. There is a strong bias toward inferred late dispersal elements of the skull and pectoral girdle and a paucity of postcranial skeletal elements at both of these sites [43,45]. Neither of these bonebeds can be distinguished from time-averaged assemblages or highly sorted assemblages with loss of early and "intermediate" dispersing elements and should not be considered mass mortality assemblages.

In depth taphonomic studies of both the Lamy amphibian quarry (Lamy) and the Rotten Hill bonebed (RH) provide points of comparison for the NK bonebed, although as noted earlier, direct comparison for the skeletal completeness is not possible due to incomplete or unpublished data. Regardless, both of these sites exhibit high degrees of skeletal sorting and loss of early dispersal elements as well highly aligned long bones, although bone alignment is less evident in RH due to the absence of a quarry map and a limited number of photos of *in situ* bones [2,3,13]. RH is found in a clast-supported matrix made up of rip-up clasts of either mudstone, siltstone, or calcrete suggesting a higher energy depositional system than that observed at both NK and Lamy [13]. Previous authors note both a lack of fine laminations in the mudstone of Lamy and a lack of ostracod or conchostracan fossils precluding the interpretation of a ponded or lacustrine depositional environment [3] and unlike the fine laminations in the mudstone of NK and the surrounding interval (see above). Rhizoliths and plant debris are common both at Lamy [3] and NK (Fig 9) but none have been reported from RH outside of mottling that may represent root traces [13]. Both Lamy and RH have been proposed to be the result of sheet flooding or crevasse splay events that rapidly transported disarticulated skeletons from the site of death [2,3,13]. No mud cracks have been observed at Lamy despite extensive excavation [3], and mud cracks have also not been observed at NK.

Site XIII and the Elkins Place (EP) bonebed could be considered two end members of metoposaurid skeletal deposits with nearly complete articulated skeletons at Site XIII and completely disarticulated and sorted skeletons at EP. Brief descriptions of the taphonomy of Site XIII have been provided in previous studies [3,13], but an overall assessment of the taphonomy of this site has not been done. In addition to abundant articulated skeletons, some details of the communications provided by [3] suggest that Site XIII represents a "drying pond" scenario as proposed by Romer [1] including: (1) large individuals were concentrated at the center of the deposit with small individuals forming a ring around them, (2) there is no "imbrication" of bones, (3) there are no mud cracks in the bonebed layer, but they are present in the same layer surrounding the quarry, and (4) no calcrete pebbles are present in the bonebed. Two of these criteria (1 and 3) require complete excavation of a site for proper comparison, and thus are not comparable with NK in its current state of excavation (i.e., the limits

of the NK bonebed have not been found). However, the lack of $B_k$ horizons (e.g., absence of calcrete pebbles) at both sites suggests some similarity in depositional regime, specifically a relatively consistent degree of sediment saturation.

At the EP site, the concentration of associated vertebrate material, the overrepresentation of late dispersal elements (Voorhies group III), and the lack of transport induced modifications (e.g., [82]) is suggestive of an autochthonous assemblage that has been winnowed or an allochthonous assemblage that underwent rapid transport. Similarly, the presence of disarticulated denticulate palatal plates in one skull from EP (fig 16: [7]) suggests either these plates were trapped in place under a skull unmoved from the site of death or the entire skull was transported with soft tissue intact. Considering the inferred seasonality during Camp Springs deposition [70], the specimen was likely a desiccated and fragmented carcass. Desiccation or draping of the bone-bearing palatal integument would have allowed for the retention of the denticulate plates leading up to final deposition where these diminutive elements were ultimately preserved in place within the interpterygoid vacuity (e.g., [119]). Along with our observations of matrix-supported clasts associated with EP, these data do not exclude the possibility of a debris flow-hosted bonebed [102,109]. However, in the absence of a quarry map and other firsthand sedimentological and stratigraphic data, these observations remain inconclusive at this time.

## Implications for the taphonomy of metoposaurids and other large temnospondyls

The skeletal composition of VDG previously used for temnospondyls have varied [2,3,13,21,27]. Ultimately, each version of VDG applied to temnospondyls ([21] (unmodified VDG); [2,3,13] (metoposaurid modified VDG); [27] (capitosaurid modified VDG)) result in similar enrichment in late dispersal elements in hydrodynamically sorted deposits despite different underlying assumptions in dispersal potential of specific elements. Regardless of which version of VDG is implemented, the overall trends are comparable (Fig 4D). Further interrogation of subtle differences in depositional regimes is possible using the framework we have presented through the inclusion of additional bonebeds coupled with actualistic studies.

In the current absence of flume experiments on temnospondyl skeletons, freshwater turtles or crocodilians may provide a much needed modern analog to better understand the taphonomy of aquatic temnospondyls (mostly stereospondyls). Bones of freshwater turtles subjected to flow in flume experiments displayed an array of preferred orientations which broadly fall into two categories: (1) plate-like elements (e.g., stereospondyl crania and pectoral girdle elements), vertebrae, and skulls showed no preferred orientation, (2) while limb bones and ribs tended to orient with the long axis parallel to flow direction [80]. Bones of alligators in flume experiments demonstrated a trend in which the interaction of height and weight of each element correlated well with dispersal potential, and in general, tall and light elements were most likely to disperse and flat and heavy objects were least likely to disperse with an array of possible morphologies in between [28]. In addition, some aquatic turtles, metoposaurids, and other stereospondyls exhibit osteosclerosis or a skeletally-variable increase in bone density for buoyancy regulation that could increase competent velocity [75,107]. These complex interactions between morphology, bone density, and dispersal potential serve to further highlight the need to conduct actualistic experiments to better understand the underlying processes in the dispersal of temnospondyl bones.

## Biochronological implications

Nearly all previous reports of vertebrate fossils from the Popo Agie Fm were restricted to the lower carbonate unit and the ocher unit with little known from the intervening purple unit.

The Nobby Knob locality from the purple unit yields the stratigraphically lowest diagnostic metoposaurid material and the first occurrence of *Buettnererpeton bakeri* from the Popo Agie Formation (Fig 5). *Buettnererpeton* was previously only known from the Camp Springs Conglomerate and the lower Cooper Canyon Formation in Texas [7,120] and the Wolfville Formation in Nova Scotia [121]. The only metoposaurid previously reported from the Popo Agie Fm is *Anaschisma browni* from the upper Popo Agie Fm or the lower-upper transitional zone [43,45]. *Buettnererpeton bakeri* is consistently found below the lowest known occurrence of *A. browni* in North American Upper Triassic strata now including the Popo Agie Formation with no reported stratigraphic overlap [7,120,121]. This succession of *Buettnererpeton–Anaschisma* is also present in the lower Dockum Group [7,120] possibly including the Otis Chalk quarries (pers. obsv. AMK; [15]).

The NK locality also yields the lowest known occurrence of a phytosaur in the Popo Agie Formation, but this specimen cannot be referred beyond Parasuchidae indet. (Fig 8). A previous report of phytosaur material from the "unnamed red beds" or lower carbonate unit in eastern Wyoming [122] has not been confirmed despite exhaustive study of material from the Clark Locality [58]. The presence of parasuchids in the Popo Agie Fm has been used to correlate it to the Otischalkian holochronozone of other Upper Triassic strata of the western USA, but the duration of Popo Agie deposition is unclear due to a lack of leptosuchomorph phytosaurs that define the succeeding Adamanian holochronozone [40,71]. The absence of leptosuchomorph phytosaurs in the Popo Agie Fm renders the whole of the Popo Agie a topless teilzone which precludes definitive correlation of estimated holochronozone boundaries in the lower Popo Agie Fm and the lower Dockum Group in the current biostratigraphic framework [71]. The base of the Otischalkian teilzone can confidently be extended to ~ 5 m above the base of the local purple unit in the Popo Agie Fm due to the presence of a partial parasuchid phytosaur mandible at this level (Fig 8). Others have mentioned the presence of "phytosaur" remains in the lower Popo Agie Fm of Wyoming (lower carbonate unit: [94]) and northeastern Utah (purple unit: [123]; "coarser parts": [124]), however the material referenced is either lost, never curated, or possibly not collected. Detrital zircon radioisotopic ages from the overlying ocher unit provide a definitively late Carnian age of the lower purple unit (ca. 230 Ma) and suggest a Carnian age for correlative strata in the lower Dockum Group [41]. The succession of the metoposaurids *Buettnererpeton* and *Anaschisma* may provide additional constraint, but the current taxonomy of metoposaurids in Texas, particularly the Otis Chalk quarries, is in need of revision.

## Conclusion

The Nobby Knob locality from the lower purple unit of the Popo Agie Formation preserves a dense association of a monospecific assemblage of the oldest known North American metoposaurid *Buettnererpeton bakeri* with minor additional faunal elements. We interpret the NK bonebed as a mass mortality event with attritional accumulation of elements surrounding the bonebed interval. Unlike other metoposaurid-dominated bonebeds in North America, the NK locality exhibits little evidence of hydrodynamic sorting. The combination of fine-grained sediment, fluvio-lacustrine characteristics of surrounding stratigraphy, partially articulated (or closely associated) skeletons, and preservation of miniscule elements in articulation are consistent with a low energy deposit such as an oxbow lake or a pond. The NK bonebed significantly increases the sample size of *B. bakeri* and may provide crucial information for elucidating intraspecific variation within the oldest known North American metoposaurids. The taxonomic composition of the NK locality is similar to sites from the lower Dockum Group of Texas with the occurrences of the metoposaurid *B. bakeri*, an indeterminate redfieldiid

ray-finned fish, abundant unionoid bivalves cf. *Antediplodon* sp., and an indeterminate para-suchid phytosaur improving biostratigraphic correlations of the Popo Agie Fm and the lower Dockum Group.

## Supporting information

**S1 Table. Metoposaurid fossils from the Nobby Knob locality cf. *Buettnererpeton bakeri*.** Composite elements such as crania and mandibles were separated into their constituent elements to estimate the completeness of crania and mandibles when isolated elements were present. Jacket numbers associate specimens collected as hand samples with the number of a field jacket. Field numbers represent either a field jacket number or a hand sample number; field jackets may have a "lab shorthand" indicated by a number or abbreviation in parentheses. Prep lab numbers are purely for bookkeeping purposes of this study, and the catalog numbers are the numbers of record (which record all numbers used for a given specimen throughout collection history).
(CSV)

**S2 Table. Metoposaurid fossils from the Elkins Place bonebed–the type locality of *Buettnererpeton bakeri*.** Composite elements such as crania and mandibles were separated into their constituent elements to estimate the completeness of crania and mandibles when isolated elements were present.
(CSV)

**S3 Table. Metoposaurid fossils from Site XIII–the type locality of *Dutuitosaurus ouazzoui*.** Composite elements such as crania and mandibles were separated into their constituent elements to estimate the completeness of crania and mandibles when isolated elements were present.
(CSV)

**S1 File. Supplemental figures and extended methods for skeletal sorting and Scanning Electron Microscopy (SEM).**
(DOCX)

**S2 File. R script for azimuthal analyses and plotting of skeletal sorting and Voorhies groups.**
(TXT)

## Acknowledgments

We thank C. Eaton and A. Goncalves for assistance with the UW Geology Museum collection and use of the collection space and equipment. We thank B. Wathen for use of a photo microscope and W. Schneider for use of a scanning electron microscope. We thank R. Rofkar for helpful discussion on place names in Wyoming. We thank J. Liu for handling an earlier version of this manuscript and S.G. Lucas and R.M.H. Smith for their constructive reviews. Our most sincere gratitude to the UWGM field crews (2014–2019) and UWGM fossil preparation lab volunteers too numerous to name who have made the study of the Nobby Knob material possible. We acknowledge the Eastern Shoshone people and their stewardship of these lands to which they have belonged since time immemorial. We also acknowledge the violation of the sovereignty of the former Shoshone Reservation (now Wind River Reservation) by western researchers whose resultant museum-based specimens we have referenced. The specimens housed at UWGM were collected under BLM permits PA16–WY–252 and PA16–WY–254 granted to DML.

## Author contributions

**Conceptualization:** Aaron M. Kufner, Max E. Deckman, Hannah R. Miller, Calvin So, David M. Lovelace.

**Data curation:** Aaron M. Kufner, Hannah R. Miller.

**Formal analysis:** Aaron M. Kufner.

**Funding acquisition:** David M. Lovelace.

**Investigation:** Aaron M. Kufner, Max E. Deckman, Calvin So.

**Methodology:** Aaron M. Kufner.

**Project administration:** Aaron M. Kufner.

**Resources:** Brandon R. Price.

**Supervision:** David M. Lovelace.

**Visualization:** Aaron M. Kufner, Max E. Deckman, Hannah R. Miller, David M. Lovelace.

**Writing – original draft:** Aaron M. Kufner, David M. Lovelace.

**Writing – review & editing:** Aaron M. Kufner, Max E. Deckman, Hannah R. Miller, Calvin So, Brandon R. Price, David M. Lovelace.

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
