## [Decision Letter · Decision Letter 0]

20 Jan 2025

PONE-D-24-59951A new metoposaurid (Temnospondyli) bonebed from the lower Popo Agie Formation (Carnian, Triassic) and an assessment of skeletal sorting in temnospondylsPLOS ONE

Dear Dr. Kufner,

Thank you for submitting your manuscript to PLOS ONE. After careful consideration, we feel that it has merit but does not fully meet PLOS ONE’s publication criteria as it currently stands. Therefore, we invite you to submit a revised version of the manuscript that addresses the points raised during the review process.

We look forward to receiving your revised manuscript.

Kind regards,

Jun Liu

Academic Editor

PLOS ONE

Journal Requirements:

“A David B. Jones Foundation grant awarded to DML helped fund the excavation of this material.”

4. We note that Figures 1-3 in your submission contain map/satellite images which may be copyrighted. All PLOS content is published under the Creative Commons Attribution License (CC BY 4.0), which means that the manuscript, images, and Supporting Information files will be freely available online, and any third party is permitted to access, download, copy, distribute, and use these materials in any way, even commercially, with proper attribution. For these reasons, we cannot publish previously copyrighted maps or satellite images created using proprietary data, such as Google software (Google Maps, Street View, and Earth). For more information, see our copyright guidelines: http://journals.plos.org/plosone/s/licenses-and-copyright.

a. You may seek permission from the original copyright holder of Figures 1-3 to publish the content specifically under the CC BY 4.0 license. 

5. We are unable to open your Supporting Information file “Copy of S2 File.R”. Please kindly revise as necessary and re-upload.

Reviewers' comments:

Reviewer's Responses to Questions

**Comments to the Author**

1. Is the manuscript technically sound, and do the data support the conclusions?

Reviewer #1: Yes

Reviewer #2: Partly

2. Has the statistical analysis been performed appropriately and rigorously? 

Reviewer #1: Yes

Reviewer #2: N/A

3. Have the authors made all data underlying the findings in their manuscript fully available?

Reviewer #1: Yes

Reviewer #2: Yes

4. Is the manuscript presented in an intelligible fashion and written in standard English?

Reviewer #1: Yes

Reviewer #2: Yes

5. Review Comments to the Author

Reviewer #1: This manuscript is a valuable contribution to bonebed taphonomy in that it addresses the long-time problem of uncontrolled and poorly documented fieldwork done my numerous short-stay field teams in the Late Triassic strata of Wyoming most of which have not been subjected to modern detailed taphonomic analysis. For the first time the authors have recorded detailed sedimentological and taphonomc data at the site as well as collecting large in-situ blocks of the bonebed for controlled mechanical preparation to further their in-depth documentation in the laboratory.

The manuscript deals with the palaeoenvironmental setting first then the bonebed genesis - giving equal emphasis to both fields. Then coming together in the discussion to formulate an evidence-supported taphonomic pathway.

The grammar is good- I have made several suggestions for improvements for clarity.

One aspect that I found a bit unnecessary for what is essentially a taphonomic rather than a palaeobiological or taxonomic paper was the systematics section. I suggest you retain the species lists but move the descriptions of taxonomically significant anatomical features to supplementary.

Lastly I would have like to see a taphonomic pathway diagram in your discussion of your most likely scenario of biogenically-induced mass mortality. It is all there, but it just needs a bit more confidence in the interpretation of your findings.

All-in-all a very good paper for the ever growing community of bonebed specialists, and a very researchworthy paper for Plos One.

Reviewer #2: This is an important article that documents a significant Late Triassic amphibian bonebed in Wyoming and evaluates its taphonomy. It needs some revision to improve the presentation, particularly the comparisons to other Triassic amphibian bonebeds. I also don’t understand the claim that somehow the Voorhies group assignments of the metopo bones are being improved on here by skeletal census? The bones are assigned to Voorhies groups by their perceived hydrodynamic qualities, not by their relative abundance. My specific comments are keyed to Comment indicators on the ms pdf:

Spencer G. Lucas

1. Lines 82-83 What does “subjective” mean here?—all of these analyses are subjective unless bones are actually being put into flumes, yes? So, how is the approach used here any less subjective than earlier work?

2. Line 197 The very recent monograph by Rinehart et al. (2024) NMMNH Bulletin 96 should be cited as well. It is up on my ResearchGate page, free download.

3. Line 334 Given that the bonebed is in pedogenically modified mudrock, why is “channel-lag in a fluvial system” even relevant here? The Lamy bonebed is also in pedogenically modified mudrock and the non-amphibian fossils are rather similar to those in the Wyoming bonebed. So, why is there no more detailed comparison Wyoming-Lamy?

4. Line 549-609 You mean “Camp Springs” not Santa Rosa, right?

5. Comparison should be made here to the Eocyclotosaurus bonebed described by Rinehart and Lucas, who also posited it as a debris flow deposit.

6. Lines 675-676 Really? How are the differences explained? Transport must be at play, as well as degree of disarticulation. This needs discussion

7. Line 730 etc. The main weakness of this paper is it does not compare the Wyoming bonebed to the most similar amphibian bonebeds (Lamy, Rotten Hill, Eocyclotosaurus) and instead focuses on the Elkins bonebed, which has never really been analyzed taphonomically and is very different (primarily skulls, in sandstone). I suggest shortening comparison to Elkins and present some meaningful comparisons to the others, particularly Lamy, which is most similar to the Wyoming bonebed.

6. PLOS authors have the option to publish the peer review history of their article (what does this mean? ). If published, this will include your full peer review and any attached files.

**Do you want your identity to be public for this peer review?** For information about this choice, including consent withdrawal, please see our Privacy Policy .

Reviewer #1: **Yes: ** Roger MH Smith

Reviewer #2: **Yes: ** Spencer G. Lucas

---

## [Author Response · Author response to Decision Letter 1]

12 Feb 2025

PONE-D-24-59951

A new metoposaurid (Temnospondyli) bonebed from the lower Popo Agie Formation (Carnian, Triassic) and an assessment of skeletal sorting in temnospondyls

PLOS ONE

Our manuscript meets the style requirements of PLOS ONE including file names.

“A David B. Jones Foundation grant awarded to DML helped fund the excavation of this material.”

The financial disclosure should read as follows: “A David B. Jones Foundation grant awarded to DML helped fund the excavation of this material. The funders had no role in study design, data collection and analysis, decision to publish, or preparation of the manuscript.”

The ethics statement present in the online submissions form was added to the Methods section of the manuscript.

4. We note that Figures 1-3 in your submission contain map/satellite images which may be copyrighted. All PLOS content is published under the Creative Commons Attribution License (CC BY 4.0), which means that the manuscript, images, and Supporting Information files will be freely available online, and any third party is permitted to access, download, copy, distribute, and use these materials in any way, even commercially, with proper attribution. For these reasons, we cannot publish previously copyrighted maps or satellite images created using proprietary data, such as Google software (Google Maps, Street View, and Earth). For more information, see our copyright guidelines: http://journals.plos.org/plosone/s/licenses-and-copyright.

a. You may seek permission from the original copyright holder of Figures 1-3 to publish the content specifically under the CC BY 4.0 license.

Figure 1 Map data and silhouettes: we added this to the figure caption “Redfieldiid silhouette by AMK and archosauromorph silhouette by DML. Chinlestegophis (stand in for Ninumbeehan) silhouette in the public domain by T.M. Keesey. Other silhouettes used under Creative Commons Attribution 3.0 Unported https://creativecommons.org/licenses/by/3.0/ from phylopic.org. Metoposaurid silhouette after D. Bogdanov and phytosaur silhouette by S.A. Hartman. Geological map and outcrop extent of Chugwater Group data (B) from https://macrostrat.org API under a CC-BY-4.0 license. The North America silhouette and state outlines (B) based on 'USA_States_Generalized' and 'Territorial_Evolution_1867_-2003_TE_' shapefiles in the public domain (Sources: Esri; U.S. Department of Commerce, Census Bureau; U.S. Department of Commerce (DOC), National Oceanic and Atmospheric Administration (NOAA), National Ocean Service (NOS), National Geodetic Survey (NGS). The paleogeographic map outline of Pangea was modified after (64). Paleoclimate map overlay modified after (65).”

Figures 2 and 3 are entirely our own work.

5. We are unable to open your Supporting Information file “Copy of S2 File.R”. Please kindly revise as necessary and re-upload.

This file can be opened in any text editing software and can be run in R.

Reviewers' comments:

Reviewer's Responses to Questions

Comments to the Author

1. Is the manuscript technically sound, and do the data support the conclusions?

Reviewer #1: Yes

Reviewer #2: Partly

2. Has the statistical analysis been performed appropriately and rigorously?

Reviewer #1: Yes

Reviewer #2: N/A

3. Have the authors made all data underlying the findings in their manuscript fully available?

Reviewer #1: Yes

Reviewer #2: Yes

4. Is the manuscript presented in an intelligible fashion and written in standard English?

Reviewer #1: Yes

Reviewer #2: Yes

5. Review Comments to the Author

Reviewer #1: This manuscript is a valuable contribution to bonebed taphonomy in that it addresses the long-time problem of uncontrolled and poorly documented fieldwork done my numerous short-stay field teams in the Late Triassic strata of Wyoming most of which have not been subjected to modern detailed taphonomic analysis. For the first time the authors have recorded detailed sedimentological and taphonomc data at the site as well as collecting large in-situ blocks of the bonebed for controlled mechanical preparation to further their in-depth documentation in the laboratory.

The manuscript deals with the palaeoenvironmental setting first then the bonebed genesis - giving equal emphasis to both fields. Then coming together in the discussion to formulate an evidence-supported taphonomic pathway.

The grammar is good- I have made several suggestions for improvements for clarity.

We have incorporated the majority of these suggestions for clarity/grammar.

One aspect that I found a bit unnecessary for what is essentially a taphonomic rather than a palaeobiological or taxonomic paper was the systematics section. I suggest you retain the species lists but move the descriptions of taxonomically significant anatomical features to supplementary.

We agree and moved the systematic paleontology section from the main text to S1 File. We also added short descriptions of types of fossils present at NK to the “Fossil occurrences” section just before the Discussion.

Lastly I would have like to see a taphonomic pathway diagram in your discussion of your most likely scenario of biogenically-induced mass mortality. It is all there, but it just needs a bit more confidence in the interpretation of your findings.

We think this is an excellent idea and have made a figure (Fig 10) to show a summary of our interpretation of the taphonomic pathway for the NK site.

All-in-all a very good paper for the ever growing community of bonebed specialists, and a very researchworthy paper for Plos One.

Reviewer #2: This is an important article that documents a significant Late Triassic amphibian bonebed in Wyoming and evaluates its taphonomy. It needs some revision to improve the presentation, particularly the comparisons to other Triassic amphibian bonebeds. I also don’t understand the claim that somehow the Voorhies group assignments of the metopo bones are being improved on here by skeletal census? The bones are assigned to Voorhies groups by their perceived hydrodynamic qualities, not by their relative abundance. My specific comments are keyed to Comment indicators on the ms pdf:

Spencer G. Lucas

1. Lines 82-83 What does “subjective” mean here?—all of these analyses are subjective unless bones are actually being put into flumes, yes? So, how is the approach used here any less subjective than earlier work?

As stated in the sentence, the reassignment of skeletal elements to “Voorhies groups” that lack experimental data is inherently subjective. Our skeletal completeness assessment (Fig. 4A–C) simply does not assume the relative dispersal potential of any element in a temnospondyl skeleton.

2. Line 197 The very recent monograph by Rinehart et al. (2024) NMMNH Bulletin 96 should be cited as well. It is up on my ResearchGate page, free download.

The study suggested here was already cited both in the text (line 193 - reference #3) as well as in the table caption (as #3) and in the table (as Rinehart et al., 2024). However, we did note redundancy in a sentence in the caption and have fixed that.

3. Line 334 Given that the bonebed is in pedogenically modified mudrock, why is “channel-lag in a fluvial system” even relevant here? The Lamy bonebed is also in pedogenically modified mudrock and the non-amphibian fossils are rather similar to those in the Wyoming bonebed. So, why is there no more detailed comparison Wyoming-Lamy?

We agree and have specified that the alignments of long bones support no evidence for unidirectional flow in a fluvial system. Additional comparison with Lamy was made in the Discussion and also noted in Reviewer 2’s comment 7.

4. Line 549-609 You mean “Camp Springs” not Santa Rosa, right?

We have clarified proposed equivalence and correlation of the Camp Springs Conglomerate with the Santa Rosa Sandstone/Tecolotito Member of the Santa Rosa Formation in the section “Specimen accessibility, preparation, and photography” on first mention of the Camp Springs Conglomerate.

5. Comparison should be made here to the Eocyclotosaurus bonebed described by Rinehart and Lucas, who also posited it as a debris flow deposit.

This is an excellent point that we had not considered. We have noted this similarity between the EP bonebed and the Eocyclotosaurus bonebed.

6. Lines 675-676 Really? How are the differences explained? Transport must be at play, as well as degree of disarticulation. This needs discussion

We have clarified the phrasing of this sentence to emphasize that the variability of the relative abundance of intermediate dispersal elements between the three sites renders any interpretation of sorting or transport essentially meaningless. Assessing the value of the intermediate dispersal elements with respect to sorting and transport interpretation would require extensive modeling or more sampling of assemblages, both of which are beyond the scope of this study.

7. Line 730 etc. The main weakness of this paper is it does not compare the Wyoming bonebed to the most similar amphibian bonebeds (Lamy, Rotten Hill, Eocyclotosaurus) and instead focuses on the Elkins bonebed, which has never really been analyzed taphonomically and is very different (primarily skulls, in sandstone). I suggest shortening comparison to Elkins and present some meaningful comparisons to the others, particularly Lamy, which is most similar to the Wyoming bonebed.

The comparison to Elkins Place is necessary and an attempt to analyze the site taphonomically given what little data is available compared to more recently excavated metoposaurid sites. Additional comparisons to Lamy and Rotten Hill were added in the “Comparison with other metoposaurid sites” in the Discussion. As noted by the reviewer, the Eocyclotosaurus bonebed is probably in a debris flow and thus not comparable with NK, but it is comparable with EP and noted in the “Elkins Place conundrum” section.

6. PLOS authors have the option to publish the peer review history of their article (what does this mean?). If published, this will include your full peer review and any attached files.

Do you want your identity to be public for this peer review? For information about this choice, including consent withdrawal, please see our Privacy Policy.

Reviewer #1: Yes: Roger MH Smith

Reviewer #2: Yes: Spencer G. Lucas

---

## [Editor Report · Decision Letter 1]

25 Feb 2025

A new metoposaurid (Temnospondyli) bonebed from the lower Popo Agie Formation (Carnian, Triassic) and an assessment of skeletal sorting

PONE-D-24-59951R1

Dear Dr. Kufner,

We’re pleased to inform you that your manuscript has been judged scientifically suitable for publication and will be formally accepted for publication once it meets all outstanding technical requirements.

Kind regards,

Jun Liu

Academic Editor

PLOS ONE
---

## [Editor Report · Acceptance letter]

PONE-D-24-59951R1

PLOS ONE

Dear Dr. Kufner,

I'm pleased to inform you that your manuscript has been deemed suitable for publication in PLOS ONE. Congratulations! Your manuscript is now being handed over to our production team.

Kind regards,

on behalf of

Dr. Jun Liu

Academic Editor

PLOS ONE